# Petrophysical and mechanical rock property database of the Los Humeros and Acoculco geothermal fields (Mexico)

Leandra M. Weydt[1], Ángel Andrés Ramírez-Guzmán[2], Antonio Pola[2], Baptiste Lepillier[3], Juliane Kummerow[4], Giuseppe Mandrone[5], Cesare Comina[5], Paromita Deb[6], Gianluca Norini[7], Eduardo Gonzalez-Partida[8], Denis Ramón Avellán[9], José Luis Macías[10], Kristian Bär[1], Ingo Sass[1,11]

[1]Department of Geothermal Science and Technology, Technische Universität Darmstadt, Schnittspahnstraße 9, 64287 Darmstadt, Germany
[2]Escuela Nacional de Estudios Superiores – Unidad Morelia, Universidad Nacional Autónoma de México, Antigua Carretera a Pátzcuaro 8701, 58190 Morelia, Michoacán, Mexico
[3]Faculty of Civil Engineering and Geosciences, Delft University of Technology, Stevinweg 1, Delft 2628CD, Netherlands
[4]Helmholtz Centre Potsdam – GFZ Research Centre for Geosciences, Section 6.2 – Geothermal Energy Systems, Telegrafenberg, 14473 Potsdam, Germany
[5]Department of Earth Sciences, University of Torino, Via Valperga Caluso 35, 10125 Torino, Italy
[6]Institute for Applied Geophysics and Geothermal Energy, EON Energy Research Center, RWTH Aachen, Mathieustraße 10, 52074 Aachen, Germany
[7]Istituto di Geologia Ambientale e Geoingegneria, Consiglio Nazionale delle Ricerche, Via Roberto Cozzi 53, 20125 Milano, Italy
[8]Centro de Geociencias, Universidad Nacional Autónoma de México, 76230 Juriquilla, Querétaro, Mexico
[9]CONACYT – Instituto de Geofísica, Universidad Nacional Autónoma de México, Antigua Carretera a Pátzcuaro 8701, 58190 Morelia, Michoacán, Mexico
[10]Instituto de Geofísica – Unidad Michoacán, Universidad Nacional Autónoma de México, Antigua Carretera a Pátzcuaro 8701, 58190 Morelia, Michoacán, Mexico
[11]Darmstadt Graduate School of Excellence Energy Science and Engineering, Jovanka-Bontschits Straße 2, 64287 Darmstadt, Germany

*Correspondence to*: Leandra M. Weydt (weydt@geo.tu-darmstadt.de)

**Abstract.**

Petrophysical and mechanical rock properties are key parameters for the characterization of the deep subsurface in different disciplines such as geothermal heat extraction, petroleum reservoir engineering or mining. They are commonly used for the interpretation of geophysical data and the parameterization of numerical models and thus are the basis for economic reservoir assessment. However, detailed information regarding petrophysical and mechanical rock properties for each relevant target horizon are often scarce, inconsistent or distributed over multiple publications. Therefore, subsurface models are often populated with generalized or assumed values resulting in high uncertainties. Furthermore, diagenetic, metamorphic and hydrothermal processes significantly affect the physiochemical and mechanical properties often leading to a high geological variability. A sound understanding of the controlling factors is needed to identify statistical and causal relationships between the properties as a basis for a profound reservoir assessment and modeling.

Within the scope of the GEMex project (EU-H2020, GA Nr. 727550), which aims to develop new transferable exploration and exploitation approaches for enhanced and super-hot unconventional geothermal systems, a new workflow was applied to overcome the gap of knowledge of the reservoir properties. Two caldera complexes located in the northeastern Trans-Mexican Volcanic Belt - the Acoculco and Los Humeros caldera - were selected as demonstration sites.

The workflow starts with outcrop analogue and reservoir core sample studies in order to define and characterize the properties of all key units from the basement to the cap rock as well as their mineralogy and geochemistry. This allows the identification of geological heterogeneities on different scales (outcrop analysis, representative rock samples, thin sections and chemical analysis) enabling a profound reservoir property prediction.

More than 300 rock samples were taken from representative outcrops inside of the Los Humeros and Acoculco calderas, the surrounding areas and from exhumed 'fossil systems' in Las Minas and Zacatlán. Additionally, 66 core samples from 16 wells of the Los Humeros geothermal field and 8 core samples from well EAC1 of the Acoculco geothermal field were collected. Samples were analyzed for particle and bulk density, porosity, permeability, thermal conductivity, thermal diffusivity, heat capacity, as well as ultra-sonic wave velocities, magnetic susceptibility and electric resistivity. Afterwards, destructive rock mechanical tests (point load tests, uniaxial and triaxial tests) were conducted to determine tensile strength, uniaxial compressive strength, Young's modulus, Poisson's ratio, bulk modulus, shear modulus, fracture toughness, cohesion and friction angle. In addition, XRD and XRF analyses were performed on 137 samples to provide information about the mineral assemblage, bulk geochemistry and the intensity of hydrothermal alteration.

An extensive rock property database was created (Weydt et al. 2020, https://doi.org/10.25534/tudatalib-201.10), comprising 34 parameters determined on more than 2,160 plugs. More than 31,000 data entries were compiled covering volcanic, sedimentary, metamorphic and igneous rocks from different ages (Jurassic to Holocene), thus facilitating a wide field of applications regarding resource assessment, modeling and statistical analyses.

## 1 Introduction

The knowledge of petrophysical and mechanical rock properties of the deep subsurface is essential for reservoir exploration and assessment of the reservoir potential for a variety of industrial applications such as petroleum reservoir engineering, geothermal heat extraction, mining or nuclear waste disposal. The data is most commonly used for interpreting geophysical data, creating conceptual geological models or populating numerical models (Lévy et al., 2018, Scott et al., 2019, Deb et al., 2019a, 2019b, Árnason, 2020). Depending on the scale of investigation (e.g. local, regional or continental scale), highly accurate spatial predictions of relevant rock properties are required to increase the success and accuracy of reservoir operations and to reduce economic risks.

Rock formations are usually characterized by a heterogeneous internal structure, mineral composition, pore and fracture distribution resulting in a great variability of petrophysical and mechanical properties (Schön, 2015). Thereby, tectonic events, diagenetic or metamorphic processes and hydrothermal alteration significantly affect the rock properties (Pola et al., 2012;

Aretz et al., 2015; Weydt et al., 2018a; Mordensky et al., 2019; Durán et al., 2019, Heap et al., 2020), leading to a high geological heterogeneity often observed within hundreds of meters to sub-meter scales (e.g. Canet et al., 2010). Although most exploration methods or geological models are aligned to the reservoir scale, the controlling factors within the reservoir need to be understood and quantified at different scales to estimate the heterogeneity of each relevant formation and to assess the uncertainty of the input parameters for different modeling approaches. However, on the one hand, detailed information about rock properties for the relevant target formations are often not available, inconsistent or distributed over the literature. On the other hand, important metadata such as petrographic descriptions, details on sample locations and applied methods for data acquisition are missing (Bär et al., 2020). Without sufficient information, it is often not possible to evaluate and profit from existing laboratory data from specific locations or reservoir formations for future modelling approaches or studies related to similar geological settings. Consequently, most reservoir models are based on assumed or generalized data sets and local geological heterogeneities are often not considered (Mielke et al., 2015). While most studies focus on a single parameter (Clauser and Huenges, 1995) or a small set of samples, extensive data sets are required, which contain data of numerous different analyses performed on each sample in order to constrain statistical and causal relationships between the parameters (Linsel et al., 2020).

Addressing these challenges, the GEMex project (Horizon 2020; GA Nr. 727550) embedded the petrophysical and mechanical rock characterization of the target formations in a comprehensive workflow providing the basis for different modeling approaches, geophysical surveys, ongoing and future volcanological studies. The GEMex project is a European-Mexican cooperation which aims to develop new transferable exploration and exploitation approaches for enhanced (EGS) and super-hot unconventional geothermal systems (SHGS). For this purpose, the Acoculco and Los Humeros geothermal fields have been selected as demonstration sites. Both fields are linked to caldera complexes located in the north-eastern part of the Trans-Mexican Volcanic Belt (TMVB). Extensive geological, geochemical, geophysical and technical investigations were performed to improve the reservoir understanding and to facilitate future drilling operations.

Up until the beginning of the project in 2016, information on rock properties of the different geological units in the study area was scarce or not available. Previous studies focused on the investigation of reservoir core samples of both geothermal fields (Contreras et al., 1990; García-Gutiérrez and Contreras, 2007; Canet et al., 2015). However, the existing data were not sufficient for the definition and parameterization of model units within the reservoir due to the limited core material available (6 pieces for Acoculco, Canet et al., 2015) or the lack of petrographic descriptions and chemical data for individual samples (Contreras et al., 1990).

Therefore, outcrop analogue studies and reservoir core studies were performed in order to characterize all relevant key units from the basement to the cap rock (Weydt et al., 2018b, Bär and Weydt, 2019). Geological heterogeneities were investigated on different scales: 1) macroscale (outcrops), 2) mesoscale (rock samples) and 3) microscale (thin section and chemical analysis). Analogue studies of the geological units exposed in outcrops around the investigated geothermal fields offer a cost-effective opportunity to investigate and correlate facies, diagenetic and metamorphic processes and lithofacies-related rock

properties from outcrops down to the subsurface (Howell et al., 2014). The definition of thermo-facies units (Sass and Götz, 2012) and the quantification of uncertainties for each parameter enable a reliable prediction of rock properties in the subsurface. A comprehensive database was developed including petrophysical, thermophyscial, magnetic, electric, dynamic and static mechanical properties combined with chemical and mineralogical data. In total 34 parameters were determined on more than 2160 plugs retrieved from 306 outcrop samples from both caldera complexes and 66 reservoir core samples of the Los Humeros geothermal field as well as 8 core samples of the Acoculco geothermal field covering volcanic, sedimentary, metamorphic and igneous rocks from Jurassic to Holocene age. Here, we present the workflow and current status of the GEMex rock property database (Weydt et al. 2020, https://doi.org/10.25534/tudatalib-201.10). This data provide the basis for ongoing research in the study area, but also facilitates a wide field of applications in different disciplines, for e.g. a first assessment of the subsurface properties at early exploration stages (Bär et al., 2020), different modeling approaches, geostatistical and stochastical analyses or for the validation of different measurement methods.

## 1.1 GEMex project framework and sampling

The geothermal system in Los Humeros is steam dominated and under production since 1990, operated by Comisión Federal de Electricidad (CFE). With a production of 94.8 MWe in 2018 it is the third largest geothermal field in Mexico (Romo-Jones et al., 2019) with 65 wells drilled so far, of which 28 are productive and five are used as injection wells. With temperatures above 380 °C encountered below 2 km depth in the northern part of the field, the Los Humeros caldera complex was characterized as a suitable target for the development of a SHGS within GEMex. In Acoculco two exploration wells have been drilled up to now, which encountered temperatures of approximately 300°C at a depth of about 2 km (Canet et al., 2015). Although a well-developed fracture network exists within the area, both wells were dry (López-Hernández et al., 2009). Thus, the GEMex project aims to develop a deep EGS in Acoculco in order to connect the existing wells to proximal fluid bearing fracture zones.

The project comprises a multidisciplinary approach based on three milestones which are 1) resource assessment, 2) reservoir characterization and 3) concepts for site development (Jolie et al., 2018). The first milestone focused on a comprehensive understanding of structural-controlled permeability and the fluid flow in the reservoir including extensive field work regarding stratigraphy and structural geology, fracture distribution, hydrological and geochemical studies of natural springs, comprehensive soil-gas studies (e.g. $CO_2$ flux, Jentsch et al., 2020) and airborne thermal imaging. The second milestone includes several geophysical surveys (e.g. passive and active seismic, gravity and magnetotelluric surveys) to characterize active faults and to identify deep structures. In addition, extensive sampling campaigns were conducted for petrophysical, rock mechanical, chemical and mineralogical investigations of the key lithologies in the study area. Resulting data and models of all work groups are being combined in integrated reservoir models at a local, regional and superregional scale. The third milestone includes the investigation of transferable concepts for developing EGS and the utilization of SGHS, the identification

of suitable materials and well designs, which can resist high temperatures and corrosive fluids in the reservoir and the
determination of possible drill pathways along with a comprehensive risk assessment and management.

The work presented in this study is part of milestone 2 (reservoir characterization) and focuses on the mineralogical, petrophysical and mechanical rock characterization of both geothermal systems. Several joint field campaigns with Mexican and European partners were conducted in order to cover and sample all relevant geological key units from the basement to the cap rock. In this context, work groups with different areas of expertise worked together in a joint approach (Fig. 1). Thus,
structural geologists worked together with volcanologists, petrologists and petrophysicists on the same outcrops to e.g. combine results of fracture pattern characterization and rock property analysis obtained from the same outcrops in a numerical fluid flow model (Lepillier et al., 2019). Likewise, samples for detailed mineralogical investigations were collected together with samples for petrophysical experiments. Over 300 representative samples were collected from more than 140 outcrops inside the caldera complexes and in the surrounding area. In addition to outcrop analysis in the Acoculco and Los Humeros
areas, particular attention was paid to the exhumed systems Zacatlán (east of Acoculco) and Las Minas (east of Los Humeros), where all units from the cap rock to the basement are exposed. These so called 'fossil systems' serve as proxies for the active geothermal fields and help to understand the fluid flow and mineralization processes in the 'active' geothermal reservoirs under discussion. Whenever possible, each geological unit was sampled several times at different outcrop locations to cover the unit's heterogeneity and only samples with an overall fresh appearance unaffected by weathering were considered.
Hydrothermal alteration of different intensities were observed in some outcrops in close proximity to fault zones and dykes. In these cases, hydrothermally altered samples were deliberately collected to analyse the effect of these processes on the rock properties. Besides analyzing outcrops and outcrop samples, CFE granted extensive sampling of wellbore core material of both geothermal fields at the CFE camp in Los Humeros. In total 66 samples drilled from 37 core sections covering 16 wells drilled in Los Humeros and 8 core samples drilled from 6 core sections from well EAC1 of the Acoculco geothermal field were
obtained. All samples were directly drilled within the field or sent as boulders to Europe or the Mexican institutes and subsequently distributed between the partners. This approach ensures that further work in the project, such as long-term flow experiments (Kummerow et al., 2020), high T/P experiments, hydraulic fracture experiments (Deb et al., 2019c), detailed mineralogical analyses (thin section and scattered electron microscope, Lacinska et al., 2020), isotope analyses or dating (Kozdrój et al., 2019) can be directly correlated with the results presented in this study. Furthermore, some parameters of the
same sample set were analyzed by multiple institutes to compare and validate different analytical approaches.

## 2 Geological setting

The Acoculco and Los Humeros caldera complexes are located in the north eastern part of the Trans-Mexican Volcanic Belt (TMVB), 125 and 180 km east of Mexico City, respectively. The E-W trending TMVB is a ~ 1000 km long calc-alkaline arc which is directly linked to the subduction of the Rivera and Cocos plates beneath the North American plate along the Middle-
American Trench (Ferrari et al., 2012, Macías et al., 2012, Avellán et al., 2018). The volcanic complexes are located over a

~50 km thick continental crust (Pérez-Campos et al., 2008) and are situated ~ 100 km north of the Popocatépetl and Pico de Orizaba volcanoes, which define the most active front of the TMVB in central-eastern Mexico (Ferrari et al., 2012, Macías et al., 2012; Avellán et al., 2020).

Both volcanic complexes are emplaced on intensively folded Mesozoic sedimentary rocks (Mexican fold and thrust belt, Fitz-Díaz et al., 2017) belonging to the Sierra Madre Oriental comprising Jurassic sandstones, shales, hydrocarbon-rich limestones and dolomites overlain by Cretaceous limestones and shales (López-Hernández et al., 2009, Fitz-Díaz et al., 2017). The regional tectonic setting is characterized by Late Cretaceous-Eocene NW-SE striking thrusts and folds and subordinate NE-striking normal faults that are associated to an Eocene-Pliocene extensional deformation phase (Norini et al., 2019). Oligocene to Miocene granitic and syenitic plutons as well as andesitic and basaltic dykes intruded into the sedimentary sequences, leading to local metamorphism of marble, hornfels and skarn (Ferriz and Mahood, 1984, Fuentes-Guzmán et al., 2020). The sedimentary basement is exposed east and southeast of the Acoculco caldera close to Chignahuapan and Zacatlán as well as in the surroundings of the Los Humeros caldera. Furthermore, it was also cut at different depth levels in drill cores in both geothermal fields (López-Hernández, et al., 2009; Carrasco-Núñez et al., 2017a). The granitic plutons are spread over the study area and new aeromagnetic data of the Acoculco caldera constrain the occurrence of at least four intrusive bodies hosted in the Cretaceous limestones at > 1 km depth. Those were interpreted as a series of horizontal mafic intrusions providing the energy to maintain the geothermal field (Avellán et al., 2020).

The Acoculco caldera complex has an 18 km x 16 km semi-circular shape (Avellán et al., 2018) and predominantly comprises Pliocene to Pleistocene basaltic to rhyolitic lavas, domes, cinder cones and ignimbrites. The caldera complex sits on an intersecting NE-SW and NW-SE fault system creating an orthogonal arrangement of grabens, half-grabens and horsts (García-Palomo et al., 2002, 2018). Thereby the regional tectonic regime strongly affected the local tectonic behavior and structural deformation of the caldera (Sosa-Ceballos et al., 2018). The Acoculco caldera is located on the NE-SW Rosario-Acoculco horst and was built on top of Cretaceous limestones, the Zacatlán basaltic plateau (so far undated) as well as Miocene and Pliocene lavas and domes related to the regional volcanism of the TMVB (Avellán et al., 2018, 2020). Thereby the pre-caldera lavas and scoria cones exposed north and northeast of the Acoculco caldera complex were related to the Apan-Tezontepec Volcanic Field (Miocene and Pliocene), whereas Miocene andesitic and dacitic lavas are exposed west of the Acoculco caldera complex. Magmatic activity of the Acoculco caldera can be divided into five different eruptive phases, including recent deposits and hydrothermal altered areas inside the caldera (Avellán et al., 2018). It began with the emplacement of the Acoculco ignimbrite (~2.7 Ma; $^{40}Ar/^{39}Ar$), followed by several early- (~2.6 – 2.1 Ma) and late-post caldera (~2.0 – < 0.016 Ma) volcanic events producing basaltic to trachyandesitic and rhyolitic lava flows restricted within the caldera and rhyolitic lava domes, scoria cones and two ignimbrites that predominantly migrated to the caldera rim and periphery, respectively. The extra-caldera volcanism (2.4 – 0.19 Ma) comprises several basaltic trachyandesitic to basaltic andesitic lavas and scoria cones, related to the volcanism of the Apan-Tezontepec Volcanic Field. Products of the extra-caldera volcanism are interbedded with the lavas of the Acoculco caldera complex. It has to be emphasized that recent studies (Avellán et al., 2018, 2020) are not in

line with previous volcanological studies performed by López-Hernández et al., (2009). In the study conducted by López-Hernández et al., (2009), the authors concluded that the Acoculco caldera (1.7 – 0.24 Ma) is nested within the older and larger Tulancingo caldera (~3.0 – 2.7 Ma) forming the so called Tulancingo-Acoculco caldera complex and that a third volcanic episode (1.8 – 0.2 Ma) occurred, which was related to monogenetic volcanism without a caldera collapse.

The younger Los Humeros caldera is the largest active caldera of the TMVB with a 21 km x 15 km irregular shape and comprises predominantly Pleistocene to Holocene basaltic andesitic to rhyolitic volcanic rocks (Carrasco-Núñez et al., 2018, Norini et al., 2019). The oldest volcanic activity in this area is represented by a thick sequence of Miocene andesites, dacites and basaltic lava flows of the Cuyoaco and Alseseca andesite unit (~10.5 Ma, Yáñez and García, 1982), and Pliocene to Pleistocene basaltic to andesitic lavas belonging to the Teziutlán andesite unit (dated between $1.44 \pm 0.31 – 2.65 \pm 0.43$ Ma, $^{40}Ar/^{39}Ar$; Carrasco-Núñez et al., 2017a). Miocene lavas have a cumulative thickness of up to 900 m and can be related to the Cerro Grande Volcanic Complex dated between 8.9-11 Ma (Carrasco-Núñez et al., 1997; Gómez-Tuena and Carrasco-Núñez, 2000) and Teziutlán andesite lavas have a reported thickness of up to 1500 m (López-Hernández, et al., 1995). Both units are classified as 'andesitic and basaltic volcanic basement' and form the currently exploited reservoir in the subsurface of the Los Humeros geothermal field (Carrasco-Núñez et al., 2018). The beginning of the magmatic activity of the Los Humeros volcanic complex is represented by rhyolitic lavas and abundant rhyolitic domes, mainly located at the western side of the volcanic complex ($270 \pm 17$ and $693 \pm 1.9$ ka; Carrasco-Núñez et al., 2018). However, the caldera collapse itself is associated with the emplacement of the high-silica rhyolite Xáltipan ignimbrite at ~160 ka with an estimated volume of 291 km³ and a thickness of up to 880 m (Carrasco-Núñez et al., 2018; Cavazos and Carrasco- Núñez, 2020). After the emplacement of the Xaltipán ignimbrite, which caused the characteristic trap-door structure of the caldera, further explosive events lead to the deposition of thick rhyodacitic Plinian deposits called Faby Tuff (Norini et al., 2015; Carrasco-Núñez et al., 2017a). Afterwards, a second caldera forming eruption occurred at ~69 ka and is related to the Zaragoza ignimbrite emplacement forming the Los Potreros caldera within the Los Humeros caldera. The post-caldera stage is represented by rhyolitic and dacitic domes within the center of the caldera ($44.8 \pm 1.7$ ka) and basaltic to trachyandesitic lava flows ($8.9 \pm 0.03$ ka), volcaniclastic breccias and fall out deposits ($7.3 \pm 0.1$ ka) with a highly variable lateral and vertical distribution (Carrasco-Núñez et al., 2017a, 2018).

**3 Workflow**

After the samples were distributed between the partners, cylindrical cores with diameters ranging from 25 to 65 mm were drilled and subsequently cut according to the standards (ASTM D4543-19, 2019) for the required sample length whereby the irregular and rough core ends were cut to be parallel. The laboratory tests were divided into three stages 1) general petrophysical characterization including all non-destructive measurements, 2) mechanical rock characterization and 3) chemical and mineralogical characterization. Non-destructive tests included particle density, bulk density, porosity, intrinsic matrix permeability, thermal conductivity at dry and saturated conditions, thermal diffusivity at dry and saturated conditions,

P-wave velocity and S-wave velocity at dry and saturated conditions, specific heat capacity, magnetic susceptibility and electric resistivity at dry and saturated conditions. Afterwards the destructive rock mechanical tests such as Brazilian Disc test, Chevron Bend test, Point Load test, uniaxial and triaxial tests were performed to determine uniaxial compressive strength, Young's modulus, Poisson ratio, tensile strength, fracture toughness, friction angle and cohesion. Samples that were identified as suitable for destructive tests such as uniaxial or triaxial tests were grinded plane-parallel prior to analysis. Quantitative and

qualitative chemical analyses like X-ray fluorescence (XRF) and X-ray diffraction (XRD) as well as thin section analyses were performed for the petrological and geochemical characterization. Figure 3 shows the schematic laboratory workflow of TU Darmstadt.

## 4 Structure of the database and sample classification

The database is publicly available under https://doi.org/10.25534/tudatalib-201.10 (Weydt et al., 2020) and contains

petrophysical and rock mechanical properties as well as chemical data obtained by laboratory experiments within the scope of the GEMex project. This database is provided in a flat file Excel format and in .csv format to keep the handling as simple as possible. Its internal structure is based on the $P^3$– PetroPhysical Property database previously developed during the IMAGE project (Bär et al., 2020) with some project-specific modifications. The $P^3$ database's internal design comprises multiple tables for petrography, stratigraphy, quality controls, chemical analyses and petrophysical properties and follows the concept of

relational database management (Codd, 1970). As the database presented in this study is restricted to one study area, the $P^3$ structure was simplified and the sample's information is compiled in two datasheets so far. The main objective was to provide the data in a user friendly and well-structured form, allowing easy filtering and a transfer of data into other data base formats like SQL (structural query language) to easily visualize it or to implement it for modeling approaches.

The first and main datasheet comprises all analyzed petrophysical parameters and sample information (meta data) compiled

during this project. Each analyzed plug was provided with a sample ID, which acts as primary key for all records. Sample information provided in the database is explained in the following sub-sections.

The second datasheet includes all chemical data, retrieved from composite sample material and does not directly correspond to measurements on single plugs. The data is provided separately to increase the handling and readability. Here, the sample name represents the primary key which links the data to the petrophysical measurements provided in the first table.

## 4.1 Meta data

The meta data include all additional sample information from sample ID to sample dimensions and can be used for rapid filtering and precise categorizing of parameters.

Each analyzed plug or sample received a unique sample ID, which is derived from the sample name given in the field, the geothermal reservoir (LH or AC), the field trip (e. g. M17 for May 2017) and an abbreviation for the rock type (e.g. GD for

granodiorite). This classification was developed within the project due to the high number of samples collected during different

field trips. Furthermore, the sample ID provides information about the sample preparation. In hierarchical order the sample name, core name and plug name are provided. For each drilled core the sample name was complemented with C1 (= core number 1), C2, C3 and so on. Whenever the core did not meet the requirements for destructive measurements (length to diameter ratio of 2:1 or too fragile), the core was cut into plugs. The core name was then complemented with capital letters A,

B, C etc. representing the way the core was cut (Fig. 4). The implementation of this hierarchical order allows for a quick access of the parameters per plug, per core or per sample. Whenever a core was not cut into several plugs, the core and plug name are identical to avoid gaps in the database. For practical reasons the term 'plug' was used for all cylindrical samples after sample preparation (cutting and grinding) ready to be analyzed. For the reservoir core samples, the existing core names were adopted. The ID begins with the well name (e.g. H23), followed by the core number (e.g. number 2), the core section (e.g. 14, or x for

undefined) and the number of the drilled subcore (C1 or C2).

The samples were classified regarding their rock type and stratigraphic unit based on the recently published geological maps and volcanological studies conducted in Acoculco and Los Humeros (Avellán et al., 2018, 2020; Carrasco-Núñez et al., 2017a and b, 2018). Rock types were predominantly determined using macroscopic analyses complemented by thin section analyses

(whenever available). Additionally, bulk chemical analyses (XRF) were used to better characterize the volcanic rocks using the TAS classification (Le Maitre and Streckeisen, 2003). However, this classification is only applicable for unaltered sample material. The classification of the stratigraphic unit is based on the international chronostratigraphic chart of the IUGS (Cohen et al., 2013) according to international standardization. Whenever possible the local stratigraphic unit is given. The volcanological studies are still ongoing and the age of some units or areas is not yet well constrained.

Coordinates of the sampling locations are provided as latitude and longitude in decimal degrees (WGS84) and X and Y coordinates (UTM WGS84). For the reservoir core samples, the coordinates of the well heads are included. All this information is given in meters above sea level (m.a.s.l.) and represents the surface evaluation of the outcrops or the evaluation at reservoir depth for the reservoir core samples. The latter was provided in measured depth (MD) by CFE, whereby the core sample material was obtained from vertically drilled wellbores.


Furthermore, the outcrop names and field trips are documented as project internal information and enable putting this work in relation to other work conducted within the study area. Samples from six field trips are provided in the database as shown in Table 1.

The 'location' was inserted in addition to the outcrop name and sample coordinates to classify the samples according to their

sampling area, distinguishing between Acoculco, Los Humeros and the exhumed systems Las Minas and Zacatlán/San Miguel Tenango (SMT). Column 'institution' refers to the institution and authors that generated the data and indirectly links it to the applied methods described in section 5.

Based on the rock type and stratigraphic classification, the samples were related to the model units of the regional and local geological models created within the GEMex project (Calcagno et al., 2018, 2020). The regional and local model units were defined to consider the most representative geological formations in the study area, the scale of the model and the objective of the project (Calcagno et al., 2018). For Los Humeros four regional and nine local model units were defined (Fig. 5). The classification is mostly based on recent work of Carrasco-Núñez et al. (2017a, 2017b, 2018) and Norini et al. (2015, 2019) and information about formation depth, thickness and distribution provided by the CFE stratigraphic drilling profiles. Samples collected from basaltic and andesitic dykes as well as intrusive bodies in Los Humeros and Las Minas were related to the basement (G4 and U9). The classification of the local units of the reservoir core samples represents the classification used for the latest update of the local model of Los Humeros (Calcagno et al. 2020).

For the regional model of Acoculco, five units were defined (Fig. 6). All volcanic deposits were merged to one unit called AC5-Volcanites, whereas the basement rocks were split into four separate units: AC4-Limestones, AC3-Skarns, AC2-Granite and AC1-Basement. The description and stratigraphic classification is based on López-Hernández et al., (2009), Lorenzo-Púlido et al., (2010), Sosa-Ceballos et al., (2018) and Avellán et al., (2018).

As the last entities belonging to the meta data, sample descriptions and dimensions for each plug are provided. The sample description includes a brief macroscopic description and gives information about the occurrence of fractures, joints and fissures or other remarks (e.g. chert nodules or stylolites). Furthermore, the information is given whether thin sections were prepared or not. The section 'sample dimensions' includes the length, diameter (exact and drilled diameter), weight (dry and saturated) and shape of the plug. Plug shapes were inserted as a quality control and distinguish between 'ideal cylindrical plug', 'cylindrical plug with a broken edge', 'irregular shape' and 'cuboid'. This information needs to be considered, when the bulk density or volume is calculated by using the sample's dimensions. The exact sample dimensions provide the opportunity to analyze scale-dependent effects (Enge et al., 2007). Therefore, plugs with varying diameter and length were drilled and analyzed. Thus, small scale samples (25 mm in diameter) for which the bulk volume reaches the minimal representative elementary volume (REV, e.g. Ringrose and Bentley, 2015) are included.

## 4.2 Rock properties

Provided rock properties are grouped as: 1) classical petrophysical parameters such as density, porosity and permeability, 2) ultrasonic wave velocities, 3) thermal properties, 4) magnetic susceptibility, 5) electric resistivity and 6) rock mechanical parameters. The results are provided as mean values with standard deviation (whenever possible) for each plug. For thermal conductivity and thermal diffusivity the maximum and minimum values were added. In total 34 different parameters were obtained following the recommendations of international norming institutions and committees (e.g. ISRM, ASTM or DIN). Columns for specific remarks were included to provide further details whenever needed. Detailed information on methods and procedures is given in section 5.

### 4.3 Chemical analyses

The results of chemical analyses (XRF and XRD) are provided in the second datasheet of the database. This data is retrieved from composite sample material and a total of 131 samples (reservoir core samples and outcrop samples) were analyzed. The sample name acts as primary key and allows for linking of chemical data with petrophysical data. Results of the XRF analyses are presented in wt % for the major elements and in ppm for the trace elements. For both analyses (XRF and XRD) the responsible institution is added to relate the data with the applied method.

## 5 Material and methods

The following sections briefly describe the applied methods conducted by the different partners. A more extensive description for the non-destructive measurements and the field trips can be found in project reports on the GEMex web page (Bär and Weydt, 2019; http://www.gemex-h2020.eu). Sample material from TU Delft (field trip January 2017) and TU Darmstadt (field trip May 2017) were distributed to GFZ, RWTH Aachen and UNITO for non-destructive petrophysical measurements.

### 5.1 Sample preparation

Drill cores with diameters ranging from 25 to 65 mm were drilled from the outcrop samples and cut into plugs as described above. More than 2100 plugs and cores with an axial length ranging from ~ 30 to 128 mm were prepared according to international standard ASTM D4543 (2019). The short plugs (diameter: 25 to 40 mm, length: 25 to ~30 mm) were predominantly used for the non-destructive petrophysical measurements like bulk density, porosity and permeability due to the specific sample size requirements of the measurement devices. Remaining plugs were prepared to meet the requirements for the different destructive rock mechanical tests, which were conducted after the petrophysical characterization. For most of the rock mechanical tests a length to diameter ratio of 2:1 (uniaxial and triaxial tests) or 1:2 (Brazilian test) is required. Furthermore, the plane surfaces of the plugs had to be plane-parallel with a maximum angular misalignment of 0.05°.

To ensure reproducibility of the results, the plugs were measured in oven-dry conditions (105 °C for more than 24 h or 64 °C for more than 48 h) and cooled down to room temperature in a desiccator (20 °C). Microcracking or significant mass losses caused by mineralogical changes or the collapse of clay minerals during heating in the oven were not observed since the majority of the outcrop samples contain no clays and samples affected by hydrothermal or metamorphic processes contain mineral assemblages developed at higher temperatures.

In order to perform measurements at saturated conditions, the samples were evacuated in a desiccator and subsequently saturated with (de-ionized) water (TU Darmstadt and GFZ) or the samples were fully immersed in water for up to four weeks (RWTH Aachen and UNITO).

## 5.2 Non-destructive tests

At TU Darmstadt, density measurements were performed in a multi-step procedure using an AccuPyc helium pycnometer (ASTM D5550) and a GeoPyc powder pycnometer (Micromeritics GmbH, Germany, 1997, 1998, 2014), analyzing particle and bulk volume five times for each plug, respectively. Bulk density was then automatically calculated by dividing the dry weight of the plug by its measured volume. Afterwards porosities were calculated from the resulting differences in volume and represent the gas-effective porosity, also known as connected porosity. The accuracy of the method is 1.1% (Micromeritics, 1998). Porosity measurements at TU Delft and UNAM were performed also using a helium gas pycnometer (Ultrapycnometer 1000 Version 2.12 and 1200e gas pycnometer, respectively; both Quantachrome Corporation, USA) to determine the grain density (ASTM D5550, 2014), while bulk density was determined using caliper techniques according to ASTM D7263 (2016). Every plug was measured up to 20 times.

At GFZ and RWTH Aachen particle density, bulk density and porosity were determined using the triple weighing method (ISRM, 1981). This method is based on the Archimedes principle, which uses the masses of the dry and fluid-saturated samples as well as that of the sample totally immersed in the fluid to calculate the pore volume and the porosity. The mass was determined with an accuracy of $\pm 0.2$ g. Usually, the accuracy is 1.5% or better, but especially depends on the surface condition for low-porosity samples. Thus, the measurements were performed up to three times per plug. A similar approach was used at UNITO by applying caliper techniques and the dry and saturated mass of each sample for the calculation of density and porosity (ISRM, 1979). Variations in particle and bulk density between the different methods applied on the same samples in this study range between 0.3-3% (coefficient of variation) for limestones with porosities smaller than 3 % and 0.5-3.5% for pyroclastic rocks with porosities between 11 and 15%, verifying the different methods and sample saturation procedures as sufficient to obtain data with the needed accuracy.

Matrix permeability was determined on cylindrical plugs (diameter and length ranging from 25 to 40 mm and ~20 to 80 mm, respectively) with column gas permeameters constructed according to ASTM D4525 (2013) and ASTM D6539 (2013) standard at TU Darmstadt, GFZ and UNAM. The plugs were analyzed in a confined cell at constant differential pressure under steady state gas flow using at least five pore fluid pressure levels (Tanikawa and Shimamoto, 2008). Corresponding gas flow rates were measured with different flowmeters that allow for the detection of flow rates in the range between 10 to 10,000 cm³ min⁻¹. This applied method is based on Darcy's law enhanced by factors for the compressibility and viscosity of gases in order to calculate the gas permeability (Scheidegger, 1974; Jaritz, 1999). The water equivalent permeability was derived from the gas permeability after the Klinkenberg correction (Klinkenberg, 1941). At TU Darmstadt the samples were analyzed with dried compressed air at five pressure levels ranging from 1 to 3 bar and 1 MPa confining pressure (Hornung and Aigner, 2004; Filomena et al., 2014). At GFZ a confining pressure of 8.5 MPa and five pressure levels ranging between 7.5 and 35 bar were applied (operated with argon), while at UNAM the permeability was determined using a confining pressure of 2.8 MPa and also five pressure levels up to 1 MPa (operated with nitrogen). Measurement accuracy of the TU Darmstadt permeameter varies from 5% for high permeable rocks ($K > 10^{-14}$ m²) to 400% for impermeable rocks ($K < 10^{-16}$ m²) (Bär, 2012). The

recorded flow rates were tested for turbulent fluid flow according to Kushnir et al. (2018) prior to the Klinkenberg correction to ensure laminar fluid flow. A correction after Forchheimer (1901) was not required, since the corrected values were within the error range of the measurement device.

At TU Darmstadt, thermal conductivity and thermal diffusivity were measured simultaneously on oven dried and saturated plugs using a thermal conductivity scanner (Lippmann and Rauen, Germany) after Popov et al. (1999, 2016). The device consists of a sample platform and an optical scanning system that moves along the sample surfaces, including a heat emitter and three infrared sensors facilitating a continuous profile. Samples are heated up by a defined heat flow and the subsequent cooling rate is measured by the temperature sensors. Bulk thermal conductivity and thermal diffusivity were then calculated after Bär (2012) by using two reference standards. Both parameters were measured four to six times on each plug for saturated and dry conditions, respectively (two to three times on every planar surface including slight turning after every measurement to account for sample anisotropy). At RWTH Aachen, the same optical scanning method was used to determine thermal conductivity along the core axis of large cylindrical cores with a diameter of 60 and 64 mm. To ensure uniform reflection conditions, the samples were painted with black acryl paint on the planar surface (TU Darmstadt) and along the core axis (RWTH Aachen). According to Lippman and Rauen (2009), the measurement accuracy for thermal conductivity is 3% and 5% for thermal diffusivity.

Specific heat capacity was determined at TU Darmstadt using a heat-flux differential scanning calorimeter (C80, Setaram Instrumentation, 2009, France), crushed sample material was heated at a steady rate from 20 up to 200 °C within a period of 24 h. Specific heat capacities were derived from the resulting temperature curves through heat flow differences. The accuracy is 1% (Setaram Instrumentation, 2009). Volumetric heat capacity was calculated by multiplying the specific heat capacity with the associated bulk density of each sample. For direct comparison, specific heat capacity was calculated for each plug by dividing thermal conductivity by the product of bulk density and thermal diffusivity (Buntebarth, 1980).

Ultra-sonic wave velocity was measured along the sample axis with pulse generators (TU Darmstadt: UKS-D including a USG-40 pulse generator and a digital PicoScope oscilloscope from Geotron-Elektronik, 2011, Germany; UNITO: Pundit Lab, Proceq, Switzerland, ASTM D2845-08, 2008; GFZ: Panametrix HV Pulser/Receiver model 5058PR in combination with a digital oscilloscope from Agilent Technologies model DSO6012A, USA) comprising point-source transmitter-receiver transducers. Thereby, the transducers were pressed against the parallel surfaces of the samples using a contact pressure of about 1 bar. Polarized pulses at high voltage in a frequency range from 20 kHz to 1 MHz for the USG-40 and Panametrix as well as 54 kHz and 250 kHz for the Pundit Lab were generated. The transmitted signals were recorded using digital oscilloscopes and the arrival times of the P- and S-waves were picked manually and corrected for the dead time, which arises from the recording device (transducer, function generator, oscilloscope).

Bulk density, P- and S-wave velocities were used to determine dynamic elastic mechanical parameters, such as dynamic shear modulus, $G_{dyn}$, dynamic Young's modulus, $E_{dyn}$, and dynamic Poisson ratio, $\mu_{dyn}$ after Zoback (2011):

$$G_{dyn} = \frac{v_s^2}{\rho} \tag{1}$$

$$E_{dyn} = \frac{\rho v_s^2 (3v_p^2 - 4v_s^2)}{v_p^2 - v_s^2} \tag{2}$$

$$\mu_{dyn} = \frac{v_p^2 - v_s^2}{2(v_p^2 - v_s^2)} \tag{3}$$

, where $\rho$ is the bulk density [kg m$^{-3}$], $v_p$ is the compressional wave velocity [m] and $v_s$ is the shear wave velocity [m].

Additional field measurements of P-wave velocities were performed by UNITO on irregular shaped outcrop samples by using the same Pundit Lab. Proceq device along different directions on the sample surfaces in order to identify anisotropy and the effect of fractures. Measurements were conducted following ASTM D2845-08 (2008) standard requirements. At TU Darmstadt both velocities were measured four to six times on each plug at saturated and dry conditions, respectively. For analyzing the samples at saturated conditions, the samples were stored in degassed and de-ionized water to avoid desaturation. After preparing the device and measurement set-up, the samples were immediately installed between the transducers and the transmitted signals were recorded until the sample starts to desaturate. The data provided by GFZ represent average values from at least four to ten individual measurements per plug (dry and saturated conditions) and at UNITO each sample was analyzed up to 20 times in order to depict the matrix heterogeneity of the larger cores and outcrop samples. The error on P-wave velocities is 3% on average, whereas for S-wave velocities the average error is 8% or higher, due to the higher attenuation and distortion of the S-wave signals.

Electric resistivity measurements were carried out on selected cylindric plugs at GFZ and UNITO and on outcrop samples in the field. At UNITO electric resistivity measurements were performed with a purpose built square quadrupole (Syskal-Pro from Iris instruments, France) after Clement et al. (2011). These consist of a rubber jacket with four steel electrodes (2 mm diameter and 40 mm length), arranged at the edges of two perpendicular diameters of the core sample at half of its longitudinal length. Electrical resistivity measurements were performed with a current injection between two subsequent electrodes and detecting the resulting electric potential between the remaining pair of electrodes. Current and potential electrodes were progressively reversed and rotated around the sample for a total of eight different potential measurements. The sequence was repeated three times and each sample was tested in both dry and saturated (wet) conditions. Saturated conditions were reached by immersing the sample in a saline solution (with electrical conductivity equal to 1000 µS cm$^{-1}$) for 24 h. A detailed description of the measurement procedure is also included in Vagnon et al. (2019).

Electric resistivity measurements at GFZ were executed with a four-electrode layout as well using an impedance spectrometer (Zahner-Zennium electrochemical work station, Zahner Scientific Instruments, 2008, Germany), which supplied an AC voltage

with an amplitude of 200 mV via disc-shaped current electrodes to the plane-parallel faces of the sample cylinders. The sample

resistance was determined via detection of the impedance and the phase angle at distinct frequencies. Subsequently, the bulk

resistivity was calculated from the sample resistance at 1 kHz, the cross-sectional area of the sample, and the distance between

the potential electrodes that were pinned to the cylinder surface of the sample plugs. The measurements were performed on

dry and on saturated samples. Oven-dry sample cores were saturated under vacuum with a NaCl solution with electrical

conductivity equal to 1080 $\mu$S cm$^{-1}$ and equilibrated for about 24 h. Prior to the measurements the samples were jacketed with

a tight-fitting silicon sleeve to reduce the risk of desaturation. The accuracy of measurements at dry conditions is better than

3.5%. In contrast, at saturated conditions for porous samples, the error increases to a maximum of 16% if fluid evaporates or

leaks from the pore space during the measurement interval.

The formation factor, $F$, of the samples was determined after Flovenz et al. (2005) from linear plots of bulk conductivities

versus fluid conductivities at different brine concentrations, where $F$ is the reciprocal of the linear fitting lines of the data

points measured at fluid salinities varying between 0.56 – 10.42 S m$^{-1}$.

Magnetic susceptibility was analyzed using the magnetic susceptibility meter SM30 (ZH Instruments, 2008, Czech Republic),

which consists of an oscillator with a pick-up coil. An interpolating mode was applied including two air reference

measurements and one measurement directly on the sample surface. The frequency change of the oscillator is proportional to

the magnetic susceptibility of the rock sample. To ensure optimal contact of the sensor on the sample surface and to reduce

the impact of air while measuring, only the plane surfaces of the plugs were analyzed.

Furthermore, a Multi-Sensor Core Logger (MSCL) from GeoTEK (2000, Germany) was used for measurements of gamma

density, P-wave velocity, magnetic susceptibility and electrical resistivity at RWTH Aachen on whole cores with a diameter

of 60 – 64 mm. Matrix density was calculated based on attenuation of gamma rays emitted from Cesium-137, while porosity

was calculated from the density measurements. P-wave velocity was measured using P-wave transducers (receiver and

transmitter) mounted on opposite faces on the center sensor stand. A short pulse is produced at the transmitter, which

propagates perpendicular to the axis of the core and is detected by the receiver on the other side. The outer diameter of the

core is measured with an accuracy of 0.1 mm. An absolute accuracy of $\pm$ 3 ms$^{-1}$ is achievable while computing the P-wave

velocity. Magnetic susceptibility was determined using a Bartington loop sensor with a 5% calibration accuracy. The sensor

includes an oscillator circuit that generates a low intensity alternating magnetic field at 0.565 kHz.

### 5.3 Destructive tests

Simple (non-cyclic) and cyclic uniaxial tests were performed to determine the rock's unconfined compressive strength and

elastic rock mechanical properties, such as the static Young's modulus, Poisson's ratio, G-Modulus (also known as shear

modulus) and Bulk modulus. For the determination of the unconfined compressive strength (UCS) at TU Darmstadt, cylindrical

plugs with a diameter of 40 mm and a length of 80 mm were introduced into a hydraulic uniaxial press (Formtest Prüfsysteme, Germany) with a capacity of 1,000 kN and a maximum loading rate of 0.5 kN s$^{-1}$ until sample failure. The stress at this particular point represents the UCS, which was calculated according to ASTM D7012 (2014) and DIN 18141-1:2014-05:


$$UCS = \frac{F}{A} \tag{4}$$

, where $F$ is the load at failure [N] and $A$ is the cross-sectional area of the sample [mm²]. Whenever the plugs were shorter than 80 mm and did not fulfil the required 2:1 length/diameter ratio, a correction function was applied as proposed by DIN 18141-1:2014-05:

$$\sigma_{U(2)} = \frac{8 \cdot \sigma_U}{7 + 2\frac{d}{l}} \tag{5}$$

, where $\sigma_{U(2)}$ is the corrected UCS [MPa] and $\sigma_U$ the measured UCS [MPa] respectively and $d$ is the sample diameter [mm], while $l$ denotes its length [mm]. At TU Darmstadt the destructive tests using the hydraulic uniaxial press were performed 'force controlled' with a maximum loading rate of 0.5 kN s$^{-1}$. The exception form very soft or fragile samples, such as ignimbrites, pumice or intensively fractured limestones. For these samples, the loading rate was individually reduced to 0.25 or 0.1 kN s$^{-1}$ to meet the test requirements and to ensure the minimal test duration. (e.g. three minutes for UCS and tensile strength).

For the determination of the static Young's Modulus and Poisson's ratio cyclic uniaxial tests were performed on three plugs (same dimension as described above) for each sample according to DIN 18141-1:2014-05 and Mutschler (2004). In order to record the axial displacement and lateral extension of the plug, three vertical and three lateral displacement transducers (LVDT) were installed in an angle of 120° around the plug. The measurement was conducted in two cycles with the first cycle reaching 40% and the second cycle reaching 60% of the previously determined UCS from the same sample set. For intensively fractured

limestones, the maximum load of the cycles was individually reduced to 30% and 50% of the previously determined UCS, respectively, to avoid an early rock failure and a possible damage of the sensors. According to Mutschler (2004) a holding time of five minutes was set at the maximum value of each cycle. After the end of the holding time of the second cycle, the sensors were removed and the sample was loaded until failure to obtain the UCS. Using the results of the first unloading cycle, the static Young's modulus (average modulus) of each plug was calculated as the difference in stress divided by the difference

in the vertical deformation according to ASTM D 3148 (2002). Likewise, the static Poisson ratio was calculated as the ratio of lateral deformation and original diameter divided by the ratio of vertical deformation and original plug length. Subsequently, G-modulus, $G$, and Bulk modulus, $K$, were calculated after ASTM D7012 (2014):

$$G = \frac{E}{2(1+\mu)} \tag{6}$$

$$K = \frac{E}{3(1-2\mu)} \tag{7}$$

, where $E$ is the Young's modulus [N mm$^{-2}$ or MPa] and $\mu$ is the Poisson ratio [-].

Furthermore, simple uniaxial tests were performed at TU Delft and UNAM to determine UCS, static Young's modulus and static Poisson ratio using a uniaxial stress/strain device with a capacity of 500 kN and 250 kN, respectively (GDSVIS Load Frame, gds instruments, UK). Plugs with a dimension of 30 mm in diameter and a length of 75 mm drilled from marble, skarn, granodiorite and limestone samples from Las Minas were tested with a loading rate of 0.15 kN s$^{-1}$ at TU Delft, while plugs with a dimension of 53 mm in diameter and a length of ~110 mm drilled from volcanic rocks from Acoculco were analyzed at UNAM (displacement controlled with 0.05 mm min$^{-1}$). Local axial and radial strains at UNAM were measured by the GDS LVDT Local Strain Transducers while at TU Delft axial displacement was recorded using two LVDTs and radial displacement was recorded using a radial chain with LVDT sensor around the plugs. UCS, static Poisson ratio and static Young's modulus (TU Delft: tangent modulus; UNAM: secant modulus at 50% of UCS) were calculated as described above following the ASTM guidelines (ASTM D 3148; 2002).

Tensile strength of the sample material was determined at TU Darmstadt and TU Delft performing the indirect tensile test, also called Brazilian test, according to ASTM 3967 (2016) and Lepique (2008). Cylindrical plugs with a diameter of 55 mm, 40 mm (TU Darmstadt) and 30 mm (TU Delft) and a diameter/length ratio of 2:1 were loaded in a hydraulic uniaxial press by a linear distributed load until failure (diametrical compression). Afterwards the tensile strength of the plug was calculated using the following equation:

$$\sigma_t = \frac{2 \cdot F}{\pi \cdot d \cdot l} \tag{8}$$

, where $\sigma_t$ is the tensile strength [N mm$^{-2}$ or MPa], $F$ the load at failure [N], $d$ the diameter [mm] and $l$ the sample length [mm].

Fracture toughness was then calculated for granite, limestone, marble and skarn samples analyzed at TU Delft after Guo et al. (1993). In order to obtain more precise values, further chevron bend tests were performed on the same sample material at TU Delft. The tests were performed on cylindrical plugs with a length of 15 mm and a diameter of 30 mm using the uniaxial device following the methods proposed by ISRM (1988). Fracture toughness ($K_{Ic}$) of the sample material was firstly determined using a direct loading to failure (= $K_{Ic}$ at Level I) and secondly using a cyclic loading to calculate the correction of fracture toughness for non-linearity (= $K_{cIc}$ at Level II).

Additionally, Point Load Tests were performed at UNAM in order to correlate the results to the tensile and uniaxial strength as proposed by ASTM D731 (2018), respectively. The tests were performed following the ISRM 325-89 (1984) and ASTM D5731-08 (2008) guidelines using a point load device from Controls (model 0550) with a maximum capacity of 100 kN.

Therefore, cylindrical plugs with a diameter of 25 mm and a length ranging between 25 and 55 mm were jacked in a neoprene membrane during the test to confine the specimen and to avoid the fragmentation due to impacts with the ground.

Triaxial compression tests were performed on oven-dry samples at TU Darmstadt using a hydraulic triaxial press (Wille Geotechnik, Germany) with a capacity of 500 kN in order to determine friction angle ($\varphi$), cohesion (c), shear ($\tau$) und normal

stress ($\sigma_n$) of the sample material. Depending on the availability, three plugs (diameter of 55 mm, length of 110 mm) for each sample were tested using different confining pressures ($\sigma_3$) of 10, 20 and 30 MPa, respectively. According to ASTM 2664 (2004) the confining pressures and resulting vertical stresses ($\sigma_1$) were transferred into a shear stress diagram to construct the Mohr-Coulomb criterion of failure to derive cohesion (intersection with the vertical axis) and friction angle (the angle between the line and the horizontal axis). Whenever needed, the vertical stresses from UCS tests (with $\sigma_3 = 0$) were considered to

construct an additional circle in the shear stress diagram, thus enhancing the data evaluation.

## 5.4 Chemical analyses

In order to perform quantitative and qualitative chemical analyses, representative composite sample material from selected outcrop samples and the reservoir core samples were milled with a disk swing mill (Siebtechnik, Germany) for 2.5 minutes at

1000 rpm at TU Darmstadt and with a colloid mill (Mixer Mill MM301, Retsch GmbH, Germany) for about 1 minute at TU Delft to obtain a grain size of smaller than 63 μm.

XRD analyses at TU Delft and GFZ were performed using a Bruker D8 Advance diffractometer (Bruker, Karlsruhe, Germany) and the software Diffrac.EVA (TU Delft) and Match! (GFZ) for data evaluation. For XRF measurements at TU Delft a Panalytical Axios Max WD-XRF spectrometer was used and data evaluation was performed with the SuperQ5.0i/Omnian

software. In addition to the Omnian standards, many NIST SRM samples and pure compounds were used for calibration. At GFZ the XRF measurements were performed with a PANalytical AXIOS Advanced spectrometer in combination with the software Super Q. For the analysis three reference standards (basalt ZGI-BM, granite ZGI-GM, and shale ZGI-TB) were used. At TU Darmstadt, major and trace elements were analyzed with a Bruker S8Tiger 4 WD-XRF spectrometer using the Quant Express method. Accuracy is < 5% for the major elements and < 10% for the trace elements. The proposed limit of detection

ranges between 400 ppm (Na) and 10 ppm (e.g. Rb, Sr, Nb). Further XRD analyses were performed at UNITO using a Siemens D-5000 automatic X-ray diffractometer. The qualitative interpretation of the data has been realized with the software "DIFFRAC PLUS, EVA Application 7.0.0.1" (2001), by comparing the positions and intensity of the data with suitable databases (JCPDS-ICCD, ICSD, PCPDFWIN).

## 6 Status of the database

The database presented here comprises petrophysical and mechanical rock properties of outcrop samples and reservoir core samples of two caldera complexes located in the northeastern part of the TMVB. So far, the database comprises 31,982 data entries (Table 2) as a result of 34 properties determined for 2,169 plugs and rock samples (2,138 cylindrical plugs and 31 uncored samples). Thereby, destructive tests were conducted on more than 970 plugs. In addition, 133 XRF and 113 XRD analyses were performed.

In total 380 samples were analyzed covering volcanic rocks (950 plugs), sedimentary rocks (716 plugs), igneous rocks (147 plugs) and metamorphic rocks (356 plugs). Thereby, 80 outcrop samples were collected for Acoculco and 226 outcrop samples were collected for Los Humeros, resulting in 563 and 1,606 analyzed plugs and samples including the reservoir core samples, respectively. The difference between the number of collected samples for Los Humeros and Acoculco is biased due to the purposes of the different field trips and the targets of the project. The main targets for the development of deep EGS in

Acoculco and SHGS in Los Humeros are marbles and skarns (AC3 and AC2) and the pre-caldera andesites and Cretaceous limestones/marbles (G3 and G4), respectively. As the basement rocks (AC1 to AC3) are not exposed in Acoculco, the exhumed systems were used as analogues. Therefore, the main attention was paid to Las Minas where 101 samples were collected (here associated to Los Humeros). In Las Minas it is possible to investigate the igneous bodies and their metamorphic products like skarn, hornfels or marble (Fuentes-Guzmán et al., 2020) as well as some outcrops belonging to the metamorphic basement

below the Cretaceous and Jurassic units.

The samples were classified regarding their model units as shown in Fig. 7. Following this approach almost all local model units for Los Humeros were covered. For some samples a classification is not possible at this stage of the project. Ongoing volcanological studies and further dating is planned to overcome these knowledge gaps. The outcrop samples belonging to the Pre-caldera group predominantly represent the Teziutlán andesite unit (U6) and the Cuyoaco andesite unit (U8). U5 comprises

ignimbrites and pumice layers from the Xaltipán ignimbrite unit, while very recent basaltic lavas, ash fall deposits and ignimbrites collected within the Los Humeros caldera were associated to the Post-caldera group (G1). The basement comprises a wide range of different rock types. G4 includes Jurassic sandstones and limestones, Cretaceous limestones, marls and shales, Miocene granitic and granodioritic intrusive bodies and their metamorphic products marble and skarn. Regarding the regional model of Acoculco, outcrop samples from the two upper units AC5 and AC4 were collected. Thereby, the uppermost unit

comprises all volcanic deposits from the pre-caldera volcanics to the extra-caldera volcanism. Among others, samples from the Acoculco ignimbrite, Terrerillos andesite lava, Manzanito andesite or Perdernal rhyolitic lava were collected. The unit AC4 includes Jurassic limestones and sandstones and Cretaceous limestones. The reservoir core samples from well EAC1 cover ignimbrite (core 1), dacitic to rhyolitic lavas (core 2 and 3), skarn (core 4), marble (core 5) and granodiorite (core 5).

The number of measurements for each parameter resulted from the availability of measurement devices at the different

institutes, required sample size, sample preparation, test duration as well as test setup. While most of the non-destructive parameters were analyzed on each plug, more time intensive tests such as specific heat capacity measurements or XRF and

XRD analyses, were performed for each sample only (composite sample material). Likewise, rock mechanical tests are significantly more time-consuming as they require a specific sample size and sample preparation or in the case of triaxial tests a minimum number of samples to evaluate the test results. Although the total number of measurements significantly differs between some parameters, all parameters were analyzed on sample sets covering all relevant lithologies in the study area.

## 7 Discussion

### 7.1 Data availability and data application

Rock properties are commonly used for reservoir exploration, assessment and modelling. While petrophysical, dynamic and static mechanical properties are the primarily used parameters for reservoir exploration, production and stimulation scenarios (Saller and Henderson, 1998, Rybacki et al., 2013, Gan and Elsworth, 2016, Ghassemi, 2017, Qu et al., 2019, Scott et al., 2019, Bohnsack et al., 2020), thermal properties are of great importance to assess the subsurface temperature, geothermal gradient, heat transport and heat storage (Weides et al., 2013, Weides and Majorowicz, 2014, Ebigbo et al., 2016, Franco and Donatini, 2017, Nurhandoko et al., 2019, Békesi et al., 2020). Especially in active high-enthalpy hydrothermal systems, electric resistivity and magnetic susceptibility data are very useful to identify or map the cap rock and different lithologies/hydrothermally altered zones within the reservoir (Oliva-Urcia, 2011, Lévy et al., 2018, 2019), whereby high T/P and detailed mineralogical studies help to estimate rock properties at reservoir conditions (Nono, et al., 2020, Kummerow et al., 2020, Lacinska et al., 2020).

Within the scope of the GEMex project, petrophysical and rock mechanical data were used for various different purposes. Deb et al. (2019b) used petrophysical and thermophysical properties to parameterize the structural model of Los Humeros and Acoculco (Calcagno et al., 2018) for simulating the initial state of the super-hot geothermal system. Several stimulation scenarios were investigated to evaluate the potential of the basement rocks in Acoculco for the development of an EGS (Deb et al., 2019a). Based on the fracture network characterization of outcrop analogues in Las Minas and petrophysical and rock mechanical data, Lepillier et al. (2019) created FEM models to calculate the fluid flow and heat exchange of fracture-controlled reservoirs in marble, skarn and limestone as an equivalent to the deep subsurface of Acoculco. Kruszewski et al. (2021) used rock mechanical parameters together with well parameters and geophysical logs to estimate the local stress field of the Acoculco geothermal field. Current studies focus on fracture propagation models and hydraulic fracture stimulation scenarios to estimate fracture geometries. The results of the petrophysical properties and volcanological studies are being used to interpret results of electric resistivity surveys (Benediktsdóttir et al., 2020), local earthquake tomography (Toledo et al., 2020) or gravity and magnetotelluric surveys (Cornejo et al., 2020).

Compared to siliciclastic or carbonate basins used for oil and gas exploitation, the amount of petrophysical and mechanical rock property data for volcanic settings in the context of high-enthalpy geothermal systems is less documented.

So far, geothermal exploration studies in volcanic settings provided rock properties analyzed whether on outcrop (e. g. Lenhardt and Götz, 2011, Pola, 2014, Mielke et al., 2016, Heap and Kennedy, 2016, Navelot et al., 2018, Mordensky et al.,

2019, Eggertson et al., 2020) or reservoir core samples (Stimac et al., 2004, Siratovich et al., 2014; Ólavsdóttir et al., 2015, Mielke et al., 2015, Cant et al., 2018). However, this study highlights the importance of the analysis of both outcrop and reservoir core samples. The comparison of reservoir samples, exhumed systems and outcrops in the surrounding area enables the identification of the processes that occurred within the reservoir and to quantify the impact on the properties correctly.

The need for valuable input data for reservoir modelling and assessment recently led to an increased number of studies and publications (Bär et al., 2020). While several extensive national or global databases already have been developed and published for geothermal well data (National Geothermal data system NGDS, 2020, BritGeothermal, 2020, DOE Data Explorer, 2020), rock chemistry, geochronology, petrology, petrophysical data such as porosity, density or magnetic susceptibility derived from geophysical borehole data (Petlab, 2020; Sciencebase Minnesota, 2020;Georoc Mainz, 2020; Rock Properties Database British Columbia Canada, 2020; Global whole-rock geochemical database compilation in Gard et al., 2019; National Geochemical Database USGS, 2020; The North American Volcanic and Intrusive Rock Database NAVDAT in EarthChem, 2020), lithology (The new global lithological map database GLiM, Hartmann and Moosdorf, 2012), mineralogy (BRITROCKS Rock collection, 2020) and petrography (RockViewer, 2020), a comprehensive and quality-proofed collection of laboratory rock properties were just recently released by Bär et al. (2020, not considering fee-based/non-open-access databases that exist for oil and gas data like the AccuMap or GeoScout databases; IHS Markit, 2020, GeoScout, 2020). The PetroPhysical Property P³ database presented in Bär et al. (2020) collected rock property data from 316 research articles and comprises 75.573 data points of 28 different rock properties analyzed on a wide variety of lithologies worldwide. While the P³ database significantly increases the availability of standardized rock properties, it still contains a limited number of data points or parameters for each investigated area or formation. To increase the level of detail for the GEMex study area to the required spatial and stratigraphic coverage, the database presented in this manuscript contains more than 31.000 data points and 34 different parameters covering all important lithologies from the basement to the cap rock. The high number of analyzed plugs and samples enables detailed statistical and spatial geostatistical analyses on different scales (plug, sample, outcrop, formation or model unit), spatial evaluation of the results in 2D or 3D or the validation of different analytical methods. Whenever possible, all parameters were analyzed on each plug. This approach allows the identification of statistical and causal relationships between the parameters and thus, improves the accuracy of geostatistical predictions, which are crucial for upscaling or downscaling (see next section, Linsel et al., 2020). The usage of plugs with different dimensions (drilled diameter ranges from 25 to 65 mm with a length from ~12 mm to 30 cm) enables the identification of scale effects, which need to be considered for the evaluation of the dynamic mechanical properties (Bayuk and Tikhotsky, 2018). The level of detail presented in this study significantly improved the geological understanding of both geothermal systems and super-hot geothermal systems in general, but also helps to better understand the relationship between different parameters and how they are affected by different processes (e.g. fracturing or hydrothermal alteration). The database provides the basis for ongoing research in the study area, but also facilitates various applications in comparable geological settings within the TMVB or similar volcanic geothermal play types worldwide. Combined with other data sets (P³, Bär et al., 2020; or Weinert et al. (2020, in review), this data could be used to train machine

learning algorithms to develop rock property prediction tools to improve and speed up parametrization of 3D geological models in the future.

## 7.2 Data processing and upscaling

The database presented in this study includes laboratory data analyzed on core and outcrop samples (cm to dm scale defined here as mesoscale), thus representing rock matrix properties only (with small-scale or single fractures in few samples). Oven-dried samples were analyzed under ambient laboratory conditions (room temperature of ~21 °C and atmospheric pressure of 0.1 MPa) to standardize the test procedure and to ensure the comparability of the results for the different samples and rock types. Consequently, the data do not reflect in situ conditions such as high reservoir temperatures, overburden pressure,

confining pressure and fluid properties at reservoir depth. Depending on the aim and scale of future applications, the data need to be corrected for reservoir conditions and transferred to reservoir scale (macroscale). Hydraulic properties such as porosity and permeability tend to decrease with increasing stress and pressure at reservoir depth by closing fractures and compaction of the rock mass (rock compressibility; Zimmermann et al., 1986, Moosavi et al., 2014, Hatakeda et al., 2017, You et al., 2020), often also resulting in increased bulk density, heat conduction, electric resistivity and wave velocities (Horai and Susaki,

1989, Clauser and Huenges, 1995, Schön, 2015). However, the relationship between different properties related to temperature and pressure changes are complex. At higher temperatures thermal expansion of minerals can cause microfracturing, which again negatively affects thermal conductivity, ultrasonic wave velocities and rock strength (Heap et al., 2014, Vinciguerra et al., 2005), but increases hydraulic properties. Several analytical and empirical relationships and correction functions have been identified and developed in the past to transfer hydraulic (Zimmermann et al., 1986, Li et al., 2004, Zheng et al., 2015, Heap

and Kennedy, 2016), thermal (Sass et al., 1971, Zoth and Hänel, 1988, Sommerton, 1992, Vosteen and Schellschmidt, 2003, Hartmann et al., 2005, Whittington et al., 2009, Rühaak et al. 2015, Zhao et al., 2016, Merriman et al., 2018, Norden et al., 2020, Clauser, 2020), magnetic (Ohnaka. 1969, Ali and Potter, 2012, Zhang et al., 2020) electric (Shankland et al., 1997, Hatakeda et al., 2017, Kummerow and Raab, 2015, Kummerow et al., 2020, Nono et al., 2020) and mechanical properties (Mobarak and Sommerton, 1971, Vinciguerra et al., 2005, Siratovich et al., 2011, Heap et al., 2014, Hassanzadegan et al.,

2013, Vagnon et al., 2020) from laboratory to reservoir conditions. Transferring rock properties from core sample to reservoir scale is challenging and has been the focus of numerous studies in the past (Christie, 1996, Farmer, 2002, Qi and Hesketh, 2005, Khajeh, 2013). Even though computer processing capacities drastically increased over the past decades, the resolution (number of grids) and complexity of static geological models often tend to be too high to run numerical reservoir simulations, which solve complex e.g. fluid or heat flow equations. Thus, upgridding and upscaling techniques are required that retain as

much of the original structure, geometry, petrophysical characteristics and facies heterogeneity as possible to deliver the vital information needed for reservoir assessment and operation (Walia and Leaky, 2014). Existing upscaling approaches can be grouped either into direct or two-step and local or global upscaling methods (Wen and Gomez-Hernandez, 1996, Farmer, 2002). The most common upscaling techniques are simple cross correlations, (power-law) averaging (arithmetic, geometric or harmonic averaging often in combination with Monte Carlo techniques), renormalization, pressure-solver or tensor methods

and pseudofunctions (Qi and Hesketh, 2005). However, especially the first mentioned techniques tend to spatially smear out extremes within the reservoir, such as flow barriers or open fractures, and thus are not very useful for complex and heterogenous reservoirs (Ding et al., 1992, Qi and Hesketh, 2005). Geostatistical analyses and modelling using estimation algorithms (e.g. variogram analyses and kriging techniques) or sequential simulations (e.g. Gaussian simulation) have been applied to populate numerical models in geological complex and/or fractured reservoirs (Hartanato, 2004, Bourbiaux et al., 2005, Ebong et al., 2019). However, integrating geological information regarding the geometry, distribution and connectivity of faults and fractures as well as linking fracture and matrix properties and fluid flow remains challenging (multiphase and dual-porosity modelling; Bourbiaux, 2010). Since hydrothermal alteration significantly influences the matrix properties (Heap et al., 2020), estimating the size and spatial distribution of hydrothermal aureoles along fractures in active volcanic settings becomes important to improve the accuracy of the reservoir model. While upscaling of hydraulic properties with application to oil and gas reservoirs has been intensively analyzed in the past (Wen and Gomez-Hernandez, 1996, Farmer, 2002, Sánchez-Vila et al., 2006), relatively little work has been done for thermal properties (Scheibe and Yabusaki, 1998, Hartmann et al., 2005, Rühaak et al., 2015). According to Rühaak et al. (2015) upscaling thermal conductivity can be fundamentally different from upscaling hydraulic or other transport parameters in porous media and rocks. The authors found that harmonic and geometric mean upscaled values most accurately reflect local values. Rühaak et al. (2014) and Gu et al. (2017) recommend kriging with external drift (KED) to interpolate subsurface temperature and thermal conductivity, respectively.

## 7.3 Limitations with respect to modelling the Los Humeros and Acoculco geothermal systems

Besides the many advantages described above, a number of limiting factors have to be considered prior to using this data set for modeling the Los Humeros and Acoculco geothermal systems. The field work and the results of the petrophysical measurements revealed the complexity of both geothermal systems. Composition, lateral extension and distribution of the volcanic sequences are very variable within the study area. Furthermore, the basement rocks showed a high geological heterogeneity comprising several different rock types including shales, limestones, sandstones, intrusive bodies, marble and skarn. The definition of the preliminary model units is predominantly based on the local stratigraphy of the study area (Calcagno et al., 2018) and some model units comprise multiple different rock types. The results of the petro- and thermophysical properties however reveal a high variability and a wide parameter range for individual units leading to high uncertainties during modeling. For this reason, the results for each lithostratigraphic unit were weighted with respect to their relative contribution in the study area for the population of the geological model of Los Humeros (Deb et al., 2019b), which was mainly based on lithostratigraphic well descriptions provided by CFE. As this is not known in detail for every model unit, the relative contribution of each rock type was based on field observations.

The number of samples per unit strongly depended on the quality, availability and accessibility of representative outcrops in the field or reservoir core samples at the core storage. Thus, it was not possible to cover all local model units for Los Humeros.

Likewise, the number of measurements for each parameter was strongly affected by the availability of measurement devices, sample preparation and test duration. Although the data for each parameter cover all key lithologies in the study area, future work should focus on additional electric resistivity and rock mechanical tests (fracture toughness and triaxial tests) to better support the interpretation of MT/TEM/DC surveys or 3D geomechanical models. Furthermore, further research is needed on HT/HP experiments reaching supercritical conditions to better evaluate the processes within the reservoir and to transfer rock properties from laboratory to reservoir conditions of super-hot geothermal systems.

The core samples of the Los Humeros geothermal field were predominantly retrieved from the reservoir pre-caldera andesite units. They show high matrix variability due to hydrothermal alteration of different intensities, which caused significant differences regarding petrophysical and thermophysical properties compared to the equivalent outcrop samples. For about one quarter of the samples, intensive hydrothermal alteration prevents a clear identification of the original rock type and correlation to equivalent units in the outcrops. This suggests that a comprehensive identification and characterization of the hydrothermal alteration aureoles in the geothermal fields is also required for the accurate assessment and modeling of these systems (e.g. by MT sounding or other direct or indirect analyses). Current studies on the reservoir core samples including detailed petrographic analyses and ICP-MS measurements aim to provide a better sample description and classification (Weydt et al., in prep.). Only a few reservoir core samples were available representing the overlaying cap rock (Xaltipán ignimbrite) or the basement below. While the Xaltipán ignimbrite unit can be investigated in several outcrops around the Los Humeros caldera, the deeper part of the basement remains mostly unknown. The high number of collected samples in the exhumed systems and in the surrounding area of the caldera complexes greatly depicts the heterogeneity of the basement. However, the analyses of outcrops and the few reservoir core samples only cover the upper limited parts of the basement (approximately ten to hundreds of meters). Thus, in the field it is not possible to investigate the spatial extension of the intrusive bodies within the (meta-) sedimentary basement. However, Urbani et al. (2020) concluded that the recent uplift within the Los Proteros caldera was caused by multiple intrusive bodies at a very shallow depth (425 ± 170 m to < 1000 m). Likewise, in Acoculco several intrusive bodies were identified at 1000 m depth already (below ground level, Avellán et al., 2020).

Regarding the regional model of Acoculco, only rocks of the two upper units are exposed in the field. For the parameterization of the remaining units, the project emphasizes to use the exhumed system in Las Minas as an analogue. Regarding the results of the petrophysical measurements, this concept can be applied for almost all units. However, the sedimentary sequences reveal the highest variability compared to other units comprising argillaceous mudstones to dolomitic marbles. The properties of the limestones and marbles resemble the different facies, diagenetic or metamorphic overprint. In Las Minas the limestones and marbles comprise dolomite, while the reservoir core samples from Los Humeros and most of the limestones collected from the outcrops in the surrounding area of both systems represent undolomitized marine, fine grained mudstones to wackestones. In addition, the reservoir core samples from the upper part of the carbonatic basement show intensive fracturing and recrystallization as a result of the complex tectonic activity caused by caldera collapses, uplift and ascending lavas. Furthermore, the term 'skarn' has been widely used in the literature (related to the study area) without a precise description. The skarns in Las Minas commonly resemble Fe-rich ore deposits in close proximity to intrusive bodies. In contrast the units

classified as skarn within the upper parts of the geothermal reservoirs (López-Hernández, et al., 2009) rather formed due to intensive metasomatic processes caused by Ca-rich fluids migrating into the overlaying lavas. Once more, the physical properties reflect the different mineralogical composition of both skarn types.

## 8 Conclusions

Within the scope of the GEMex project, an extensive rock property database was created comprising more than 31,000 data entries covering a great variety of different rock types and lithologies of Jurassic to Holocene age. The database includes petrophysical, thermophysical, magnetic, electric and dynamic and static mechanical properties complemented by the results of XRF and XRD analyses. In total 34 properties were determined on 2,169 plugs retrieved from more than 300 outcrop samples collected from the Acoculco and Los Humeros caldera complexes, 66 reservoir core samples drilled from 37 core sections from 16 wells of the Los Humeros geothermal field and 8 core samples drilled from 6 core sections obtained from well EAC1 of the Acoculco geothermal field. The database was created in a simple and transparent format including comprehensive meta information to facilitate the application in various geoscientific disciplines worldwide.

The compiled data set allows for:

- prediction of rock properties of target formations in the subsurface at early exploration stages or in case of low data density
- assessment of the reservoir potential and estimation of economic risks and uncertainties
- population of 3D geological models (numeric thermo-hydraulic-mechanical-chemical (THMC) models)
- statistical evaluation to identify relationships between the properties and trends required for upscaling approaches and
- validation of different analytical methods.

The data and workflow presented here will improve the planning and execution of future research projects. Outcrop analyses and the characterization of petrophysical and mechanical properties of outcrop and reservoir core samples are paramount for a profound reservoir characterization and should in general be considered in future geoscientific studies to a greater extent to enable a more precise prediction of reservoir properties. Hereby, an integration of shallow geophysical and classical (scan-line etc.) or state of the art (LidaR) fracture network characterization methods has a great potential to further enhance 3D reservoir characterization.

The current structure of the database allows for an easy modification and extension. It is planned to create an outcrop catalogue of all field campaigns conducted within GEMex and to improve it by adding the results of ongoing ICP-MS and detailed petrographic analyses.

**Author contributions** This study was conducted by LMW from sample analyses, to data evaluation and compilation and writing the manuscript. AARG, AP, BL, JK, GM, CC and PD contributed to the database by providing results of rock property measurements and chemical analyses. LMW, AARG, AP, BL, GM, CC, KB, GN, EGP, DRA and JLM were involved in the sampling campaigns. Thereby GN, EGP, DRA, JLM provided the main input to the geological interpretation and sample classification. The concept of the study as well as the manuscript outline and composition were designed by LMW, KB and IS. Rock characterization was coordinated by KB, IS and AP for the European and Mexican consortium, respectively. All authors contributed to this study and reviewed the manuscript.

**Competing interests** The authors declare that they have no conflict of interest.

**Data availability** The data repository is available under https://doi.org/10.25534/tudatalib-201.10 (Weydt et al., 2020).

## Acknowledgements

We thank Ing. Miguel Angel Ramírez Montes Subgerencia de Estudios Gerencia de Proyectos Geotermoeléctricos and the Comisión Federal de Electricidad (CFE) team for providing us access to the core storage and their help during sampling and drilling the CFE core samples. We highly appreciated working at the Los Humeros camp.

Furthermore, we thank Jana Perizonius, Thomas Kramer, Maximilian Bech and Roland Knauthe (Master and Bachelor graduates at TU Darmstadt) for their contribution to this project.

Special thanks to Cord Peters, Patrick Höfler, Fatih Ekinci, Ruud Hendrikx, Gabriela Schubert, Dirk Scheuvens, Rainer Seehaus, Reimund Rossmann, Georg Wasmer, Angelo Agostino, Jessica Chicco, Chiara Colombero, Anna Ferrero, Sergio Vinciguerra and Federico Vagnon for their great support in the laboratory.

We also thank Christopher Rochelle, Domenico Liotta and his team, Víctor Hugo Garduño-Monroy† and his students, Geovanny Hernández-Avilés, Luís and Daniel González-Ruiz, Irais Franco and Gerardo Carrasco-Núñez for their support during the field campaigns. Many thanks to Caterina Bianco for providing Fig. 1c.

This project has received funding from the European Union's Horizon 2020 research and innovation programme under grant agreement No. 727550 and the Mexican Energy Sustainability Fund CONACYT-SENER, project 2015-04-68074.

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

**Table 1: Overview of the field campaigns and related work**

| No. | Field campaign | Related work |
|---|---|---|
| 1 | January 2017 | Mapping, structural and mineralogical analyses in Las Minas and Acoculco (Liotta et al., 2019; Lepillier et al., 2019) |
| 2 | March 2017 | Hydraulic fracture experiments on large blocks (Deb et al., 2019c) |
| 3 | May 2017 | Structural analyses in Los Humeros and Las Minas (Norini et al., 2019), samples for high temperature triaxial tests (Vagnon et al, 2020, Bär and Weydt, 2019), samples for long-term flow through experiments at supercritical conditions (Kummerow et al., 2020), samples for scanning electron microscopy, electron probe microanalysis, cathodoluminescence microscopy and high temperature fluid-rock reaction experiments (Lacinska et al., 2020; Bär and Weydt, 2019) |
| 4 | June 2017 | Petrophysical characterization and mechanical evolution of hydrothermal altered rocks |
| 5 | January 2018 | Mapping, structural and mineralogical analyses in Acoculco and Las Minas (Liotta et al., 2019, Lepillier et al., 2019), dating (Kozdrój et al., 2019), samples for high temperature triaxial tests (Vagnon et al, 2020, Bär and Weydt, 2019), samples for scanning electron microscopy, electron probe microanalysis, cathodoluminescence microscopy and high temperature fluid-rock reaction experiments (Lacinska et al., 2020; Bär and Weydt, 2019), samples for fluid inclusions (Ruggeri et al., 2020) |
| 6 | March 2018 | Shallow geophysical surveys, determination of mechanical properties at field scale, electrical resistivity tomography (Mandrone et al., 2020) |

**Table 2: Number of measurements for each parameter**

| Parameter | No. of measurements |
|---|---|
| Particle density | 1,878 |
| Bulk density | 1,379 |
| Porosity | 1,352 |
| Permeability | 1,052 |
| Thermal conductivity (dry) | 1,669 |
| Thermal conductivity (sat) | 1,465 |
| Thermal diffusivity (dry) | 1,617 |
| Thermal diffusivity (sat) | 1,396 |
| Specific heat capacity | 210 |
| Specific heat capacity (calculated) | 1,093 |
| Volumetric heat capacity | 210 |
| P-wave velocity (dry) | 1,819 |
| S-wave velocity (dry) | 1,753 |
| P-wave velocity (sat) | 1,416 |
| S-wave velocity (sat) | 1,375 |
| Dynamic Young's modulus (dry) | 1752 |
| Dynamic Young's modulus (sat) | 1,375 |
| Dynamic Poisson ratio (dry) | 1,736 |
| Dynamic Poisson ratio (sat) | 1,375 |
| Dynamic Shear modulus (dry) | 1,743 |
| Dynamic Shear modulus (sat) | 1,375 |
| Magnetic susceptibility | 921 |
| Electric resistivity (dry) | 31 |
| Electric resistivity (sat) | 50 |
| Formation factor | 39 |
| UCS | 465 |
| Static Young's modulus | 242 |
| Static Poisson ratio | 243 |
| Shear modulus | 209 |
| Bulk modulus | 209 |
| Tensile strength | 407 |
| Fracture toughness | 86 |
| Friction angle | 20 |
| Coehsion | 20 |
| **Total** | **31,982** |


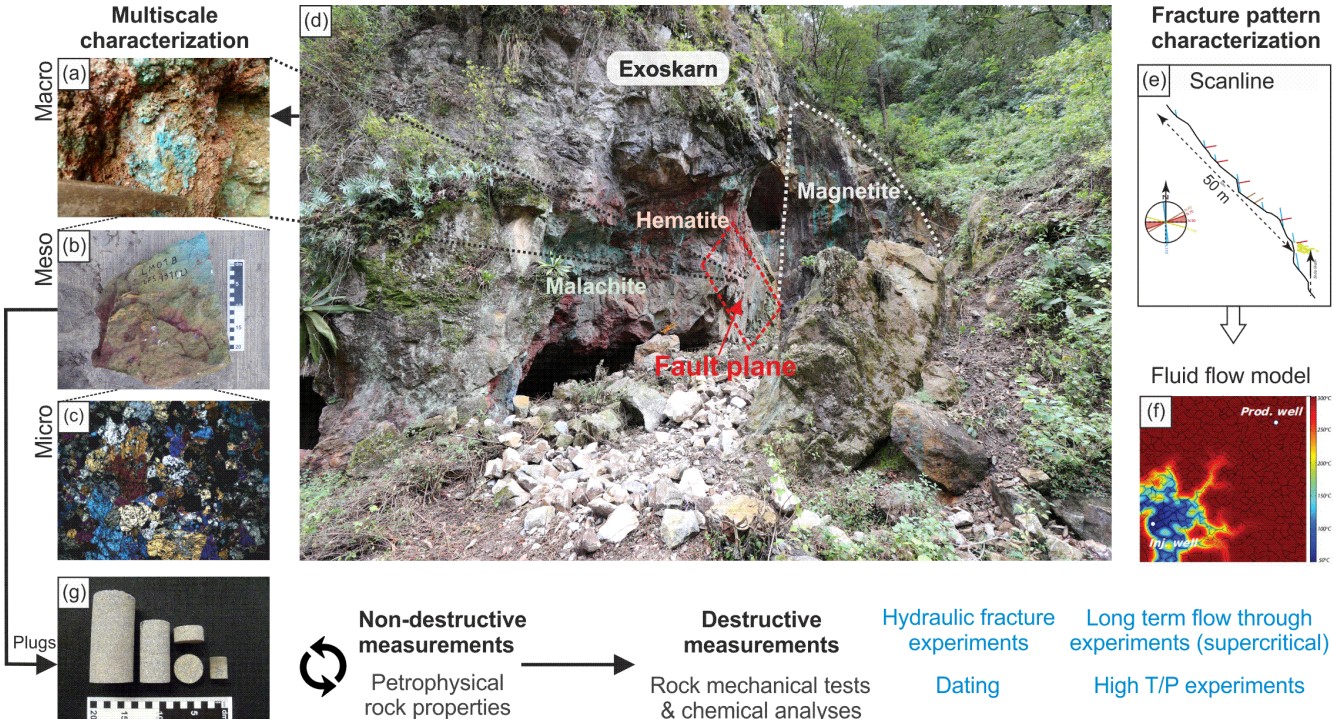

**Figure 1: Schematic workflow of the GEMex project using the example of the El Dorado mine in Las Minas (d) with view on the footwall of the present fault (photo from Maximilian Bech). The quarry exposes exoskarn in many variations. Outcrop analysis included detailed investigation of kinematic indicators, mineralogy (a) and the main fracture pattern (e) to create numerical fluid flow models (f) as presented in Lepillier et al. (2019). Rock samples were taken for lab investigation (b), geochemical and thin section analysis (c) (photo from Caterina Bianco). Cylindrical plugs were drilled from the outcrop samples (g) and distributed between the partners in order to determine rock properties, dating or highT/P experiments (the experiments marked in blue are not included in this study).**

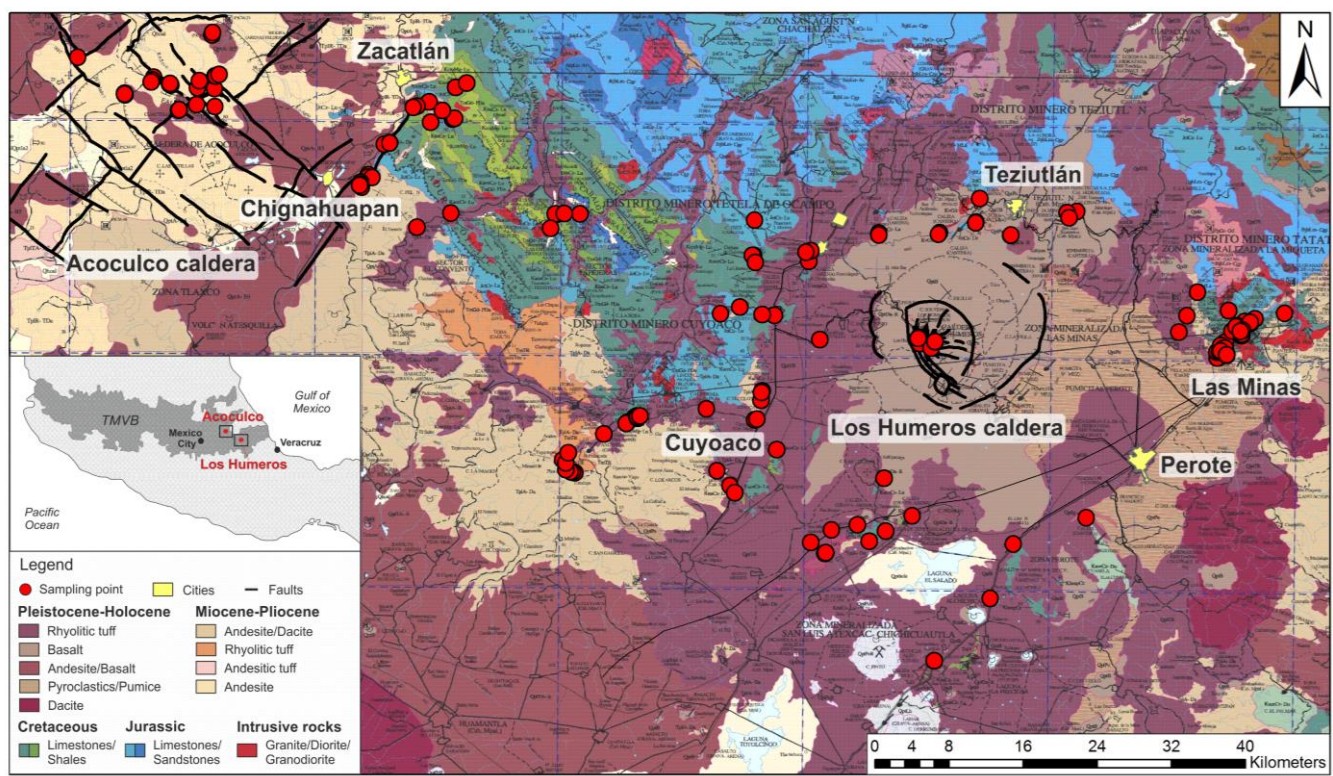

**Figure 2: Geological map of the Acoculco and Los Humeros region including the sampling points of the outcrop samples (SGM, 2002a and b). The faults were recently mapped and characterized by Liotta et al. (2019) and Norini et al. (2019).**

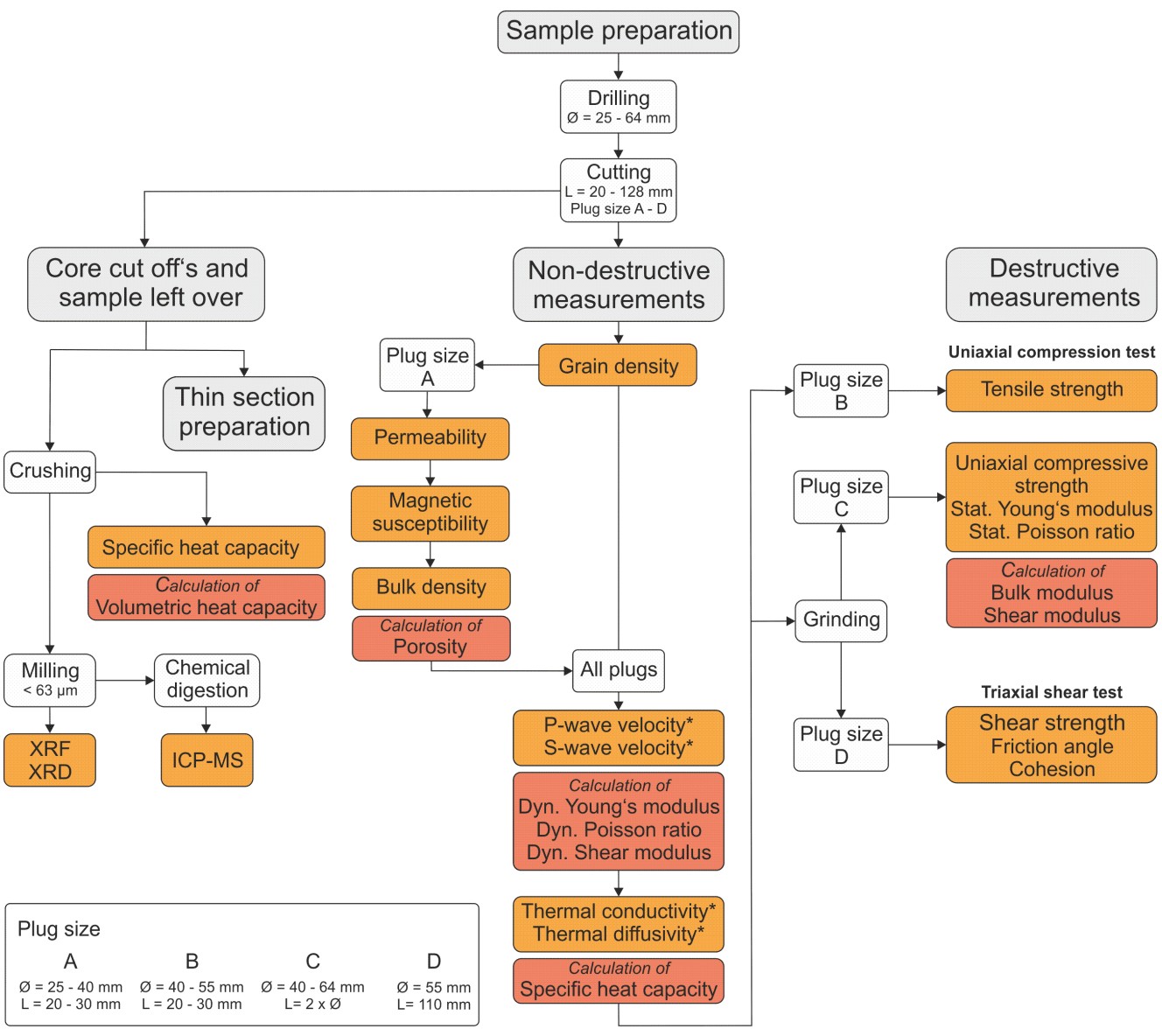


**Figure 3: Schematic work flow representing the measurement procedure at TU Darmstadt. The properties displayed in orange were determined on sample material and used to calculate those shown in red. Parameters marked with * were analyzed at dry and saturated conditions.**

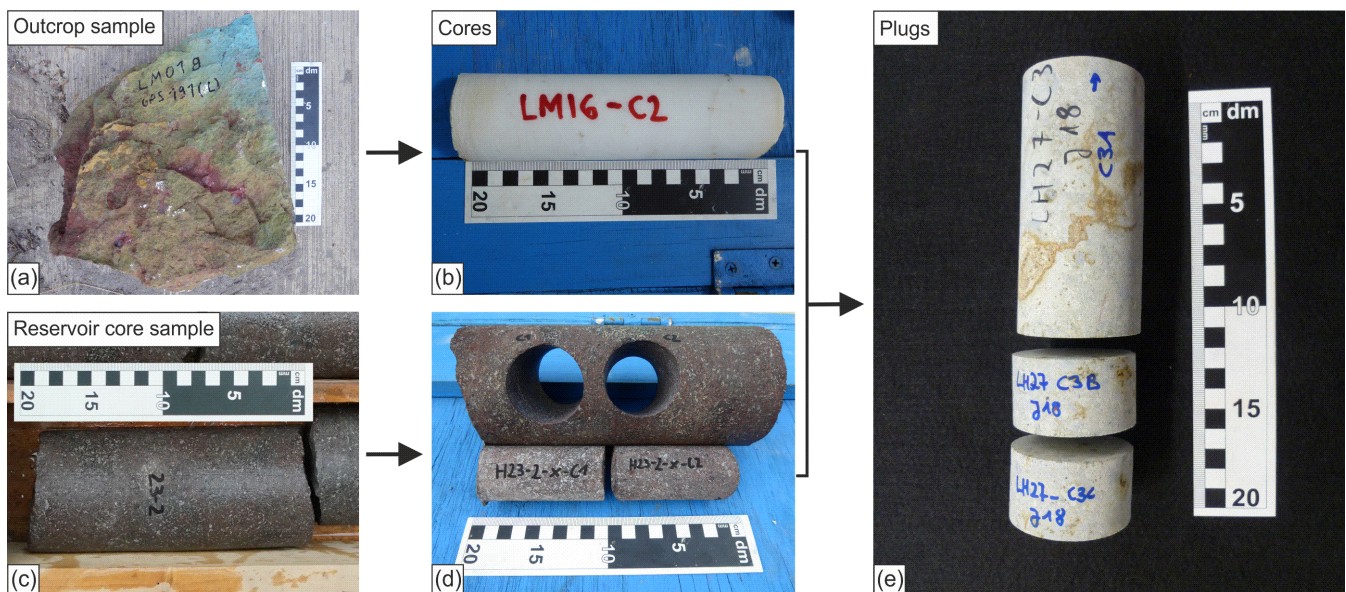

**Figure 4: Overview of the different preparation steps and sample labelling. Cores (b = various diameters, d = 40 mm in diameter) were drilled from outcrop samples (a) and reservoir core samples (c) and subsequently cut into plugs (e) to meet the individual requirements of the measurement devices. The plugs were labelled with capital letters.**

| Regional model | Local model | Rock type | Age (Ma) |
|---|---|---|---|
| G1 Post-caldera | U1 Undefined pyroclastic | Tuff, pumice and some alluvium | < 0.003 |
| | U2 Post-caldera | Rhyodacite, andesite, basaltic andesite and olivine basalt lava flows with intercalated pyroclastic deposits | 0.003-0.050 |
| G2 Caldera | U3 Los Potreros caldera | Rhyodacitic flows and Zaragoza ignimbrite | 0.069 |
| | U4 Intermediate caldera | Faby tuff with andesiti-dacitic flows, rhyolitic and obsidian domes | 0.07-0.074 |
| | U5 Los Humeros caldera | Mainly composed of Xaltipán ignimbrite with minor andesitic and rhyolitic lava | 0.165 |
| G3 Pre-caldera | U6 Upper pre- caldera | Pyroxene andesites (Teziutlán andesite unit) with mafic andesites in the basal part and/or dacites and rhyolites | 1.46-2.61 |
| | U7 Intermediate pre-caldera | Undifferentiated Rocks: Intercalation of rocks highly altered whose origin has not been defined so far | 2.62-8.8 |
| | U8 Basal pre-caldera | Hornblende andesites (Alseseca andesites and Cerro Grande volcanism) and dacites | 8.9-10.5 |
| G4 Basement | U9 Basement | Middle Miocene granite | 15.12 |
| | | Cretacic limestones, shale and minor flint | ~140 |
| | | Jurassic limestone and shale | ~190 |
| | | Paleozoic granite and schist (Teziutlán Massif) | > 251 |

**Figure 5: Regional and local model units of the 3D geological model of Los Humeros (slightly modified from Calcagno et al. 2018, 2020).**

| Regional model | Rock type | Age |
|---|---|---|
| AC5 Volcanites | Ignimbrites, dacites, rhyodacites, andesite (pre to post caldera volcanites plus extra caldera and alluvial units) | < 12.7 Ma |
| AC4 Limestones | Limestone, marbles, hornfels | Cretaceous |
| AC3 Skarns | Limestone skarns | Cretaceous |
| AC2 Granite | Hornblende granite and microgranitic dykes | Mid-Miocene |
| AC1 Basement | Phyllites | Paleozoic |

**Figure 6: Regional model units of the 3D geological model of Acoculco (slightly modified from Calcagno et al. 2018).**


(a)

| Regional model | Local model | No. of samples |
|---|---|---|
| G1 Post-caldera | U1 | 1 |
|  | U2 | 9 |
| G2 Caldera | U3 | 4 |
|  | U4 | - |
|  | U5 | 15 |
| G3 Pre-caldera | U6 | 35 |
|  | U7 | 8 |
|  | U8 | 14 |
| G4 Basement | U9 | 169 |

(b)

| Regional model | No. of samples |
|---|---|
| AC5 Volcanites | 46 |
| AC4 Limestones | 40 |
| AC3 Skarns | 1 |
| AC2 Granite | 1 |
| AC1 Basement | - |

**Figure 7: Number of collected samples (outcrop and reservoir core samples) per model unit for the regional and local models of the Los Humeros (a) and Acoculco (b) geothermal systems.**