# Peer review of "Petrophysical and mechanical rock property database of the Los Humeros and Acoculco geothermal fields (Mexico)"

_Earth System Science Data, 2020_

## Referee Comment (RC1) · Anonymous Referee #1 · 29 Jul 2020

Earth Syst. Sci. Data Discuss.,

https://doi.org/10.5194/essd-2020-139-RC1, 2020 © Author(s) 2020. This work is distributed under

the Creative Commons Attribution 4.0 License.

Received and published: 29 July 2020

This manuscript presents a rock physical property database for two geothermal fields in Mexico. The database contains a colossal volume of data. This contribution summarises the motivation for the project and the geological structure of the area, before explaining the workflow and methods used to compile the database. The manuscript is well written and logically ordered. My only major comment is that I think, as explained further below, the discussion section should include an additional paragraph that outlines the issues surrounding using laboratory-measured values in large-scale models (i.e. upscaling). I consider this manuscript suitable for publication after the following comments have been suitably addressed to the satisfaction of the editor.

Line 68: Another relevant, and recent, paper that the authors could consider citing here is Heap et al. (2020, JVGR). Heap, M. J., Gravley, D. M., Kennedy, B. M., Gilg, H. A., Bertolett, E., & Barker, S. L. (2020). Quantifying the role of hydrothermal alteration in creating geothermal and epithermal mineral resources: The Ohakuri ignimbrite (Taupo Volcanic Zone, New Zealand). Journal of Volcanology and Geothermal Research, 390, 106703.

Lines 90, 107, and 254 (and elsewhere): Data is plural.

Line 146: "Samples...were collected several times..." suggests that the same block of rock was collected several times. Suggest to reword.

Line 265: I suspect the authors mean "too friable" rather than "too brittle".

Line 329: You mean "Table 2"?

Line 350: A temperature of 105 °C might be high enough to encourage thermal microcracking or damage clays. Can the authors comment on the suitability of using this temperature? Are the authors sure the materials were not affected?

Lines 353 and 439: Can the authors comment on the effectiveness of saturating samples by leaving them submersed in water? For tight rocks, it seems doubtful that water would have penetrated thin pores/cracks. Errors resulting from incomplete saturation would influence, for example, the porosity measurements using the triple-weight method (Lines 365 and 370).

Line 360: For those unsure of the meaning of "effective porosity", I would add "i.e. connected porosity" in parentheses here.

Line 373: What was the range of plug length? Measurements on "short" samples of a homogeneous sandstone were recently shown to provide reliable permeability values, see Heap (2019). These authors argued that permeability measurements on "short" samples are reliable as long as the pore/grain/crystal size is small compared to the length/diameter of the sample. Heap, M. J. (2019). The influence of sample geometry

ESSDD
on the permeability of a porous sandstone. Geoscientific Instrumentation, Methods and Data Systems, 8(1), 55-61.

Line 377: Gas permeability measurements for high-permeability samples and/or when using high flow rates likely also require a Forchheimer correction. Did the authors check for this?

Line 381: "...at five pore fluid pressure levels..."

Line 407: Elastic wave velocities were measured parallel to the sample axis?

Line 425: Can the authors provide more information as to how the saturated velocities were measured? On samples submersed in water? Or were the samples wrapped in cling film and quickly measured to avoid desaturation?

Line 474: If the authors prefer to use "G-Modulus", I would also put "shear modulus" in parentheses to avoid any confusion.

Line 479: This should be "load at failure/maximum load" and "cross-sectional area".

Line 484: Do the authors mean here that they used a constant loading rate of 0.5 kN/s? It's not clear. Written as it is, it suggests that the loading rate was variable and that the maximum was 0.5 kN/s.

Line 485: Do the authors mean the loading rate?

Line 490: What type of sensor? Strain gauges?

Line 508: Can the authors elaborate on what they mean by "tension controlled"?

Line 519: They were loaded diametrically in compression?

Line 537: The triaxial experiments were performed on dry samples?

Line 607: See also the study by Eggertsson et al. (2020), who measured samples taken from the Krafla geothermal system in Iceland. Eggertsson, G. H., Lavallée, Y., Kendrick, J. E., & Markússon, S. H. (2020). Improving fluid flow in geothermal
reservoirs by thermal and mechanical stimulation: The case of Krafla volcano, Iceland. Journal of Volcanology and Geothermal Research, 391, 106351.

Line 669: I think the authors should include an additional paragraph(s) that states that large-scale modelling, such as fluid circulation models, require upscaled values not those measured in the laboratory. I think it would be beneficial for the reader if the authors explain the issues surrounding using laboratory-measured values in large-scale models and discuss/present existing methods typically used to upscale such values.

---

## Referee Comment (RC2) · Léa Lévy (Referee) · 16 Aug 2020

Pdf attached has better formatting.

1. General comments on the paper

The manuscript presents a unique, robust and extensive database of a large range of petrophysical, mineralogical and chemical properties of volcanic rocks from two geothermal fields in Mexico. The workflow and methods are well described. It also presents some limitations in relation to the use of data. The database is easily accessible from one unique excel file, which is (surprisingly!) only 1.7 Mo and can be

open without issues. Moreover, I find it easy to navigate and understand the database. Standard deviations of most measured parameters are clearly presented, which will be of great value for future users, especially for feeding geo-statistical analyses.

I consider this manuscript suitable for publication but I suggest a moderate revision in order to address two important changes in the paper, which are required in my opinion:

(i) More explanations about the lack of electrical measurements (50 samples versus 1000-1500 for all other properties) and the implications for statistical analyses and MT/TEM/DC surveys interpretations. MT and TEM are among the most common methods (if not the most) used in geothermal exploration, so this discussion is critical to justify the usefulness of your paper. The paper also should help the reader find ways to overcome this gap. Additionally, the use of ERT for inferring the resistivity of samples is not a state-of-the-art method and it is not clear how many of the 50 samples are inferred from ERT. Especially because 50 samples have a formation factor, which I guess you cannot obtain with only ERT measurements. More clarity is needed here.

(ii) Clearer aim and context. References and comparison to recent, similar and complementary studies are lacking. Especially to feed the discussion on how to overcome limitations of this specific database. I have suggested a few studies that I know of and consider complementary.

2. Specific comments related to the scientific content

I find the aim of this study somewhat unclear but I think it could be easily improved. In the abstract you only mention "overcome the gap of knowledge of the reservoir properties". It could be clearer, more specific and also presented with more perspective and context. This would make the paper and database immediately relevant for a large audience and allow exciting scientific discussion. It would show the usefulness of the database in a clearer manner.

To be more specific, my main question after reading the paper is: Is this database only

intended at interpreting geophysical datasets in the two corresponding areas in Mexico, for this corresponding deepEGS exploration project? Or do you see possibilities to use this database at other geothermal fields, for different geothermal exploration projects?

- If this is only for the present exploration project in Mexico, are you then suggesting that such extensive data collection be done for every geothermal system to be explored? It would be interesting to get an idea of how much resource it requires, compared to other exploration costs. Is it realistic? Are we going to need public funds for every new geothermal exploration project? Or is there a point where we will have hopefully collected enough petrophysical data and run sufficient statistical analyses, to be able to build experience from one field to the other, and even compare fields world-wide?

- If you consider that this database can be used in other contexts than in Mexico, it would be very valuable to elaborate a bit. How can a given petrophysical dataset be used to better understand reservoir behavior? To interpret geophysical data at other places?

In general, I think this is a very important and interesting discussion and your paper is a great opportunity to foster it (with a section in the discussion section?).

Regardless the answer to the question above, I think it is necessary to put this study more in context with similar studies. It is not the first time that such a massive effort is made in the context of high-enthalpy geothermal exploration. I can see that you refer to the P3 database made in the frame of IMAGE project, where the focus was on Iceland and Italy. I think it is critical to expand a bit on the differences and on the coherence between the two projects / databases. Why is this new database necessary after the one in the IMAGE project? How are they complementary? What results from IMAGE have convinced you that making such a database was useful? This would be a useful addition at lines 80-87.

There is also a range of (recent) studies that already use extensive and relevant petro-physical dataset to interpret/calibrate their conceptual models. I think that referring

to them would give more weight to your paper, by emphasizing how useful it is to have petrophysical data to calibrate geophysical data. You can stay in the field of geothermal-related studies or even extend a bit broader to sedimentary context. I have listed below a few studies that I find particularly relevant.

a. Suggestion of additional references

References to similar studies, complementary data, and successful application of petrophysical calibration of geophysical data are missing in the introduction and/or discussion. Suggestions below.

"Imaging the magmatic system beneath the Krafla geothermal field, Iceland: A new 3-D electrical resistivity model from inversion of magnetotelluric data"

–> Interpret MT inversions at geothermal fields using petrophysical calibration (especially temperature dependent measurements).

"New Conceptual Model for the Magma-Hydrothermal-Tectonic System of Krafla, NE Iceland" https://www.mdpi.com/2076-3263/10/1/34

–> Shows how conceptual models are regularly updated in light of new petrophysical understandings

Study related to both IMAGE and GEMEx projects: "Electrical resistivity tomography and time-domain induced polarization field investigations of geothermal areas at Krafla, Iceland: comparison to borehole and laboratory frequency-domain electrical observations" https://academic.oup.com/gji/article-abstract/218/3/1469/5497301

–> Interpret DC/IP inversions based on petrophysical measurements on core samples at the exact same site. Discussion on upscaling with in particular comparison of samples to borehole logging and analyses of in-situ versus laboratory temperature differences.

"A      probabilistic      geologic      model      of      the      Krafla      geothermal      system constrained by gravimetric data" https://geothermal-energy-journal.springeropen.com/articles/10.1186/s40517-019-0143-6

–> Statistical analysis of the link lithology versus density, and use to interpret gravity data. Could be cited around l. 79.

"Subsurface imaging of water electrical conductivity, hydraulic permeability and lithology at contaminated sites by induced polarization" https://academic.oup.com/gji/article/213/2/770/4816733

–> Lithology and permeability characterization using petrophysical calibration based on extensive laboratory database measured at a different area (German sediments in the laboratory used to interpret geophysics in Denmark).

You are saying in the introduction that data are distributed in different places (l.72-79), which makes their use complicated. But if this database is intended to be used in a more general manner than just in this project in Mexico, then there needs to be a (short) section on other similar database and how they can be combined. It could be in the discussion as well. I would also add references, either in the introduction (near l.72-79) or in the discussion, to data collection presented in separated papers or PhD thesis, provided that the data collection is significantly large and well-presented and contains consistent data to be comparable to your database, of course. That way the reader will know where to find complementary information, if he needs, e.g. in the IMAGE database or in other articles. A few suggestions below.

"Modification of the magnetic mineralogy in basalts due to fluid–rock interactions in a high-temperature geothermal system (Krafla, Iceland)" (see Table A1)

https://academic.oup.com/gji/article/186/1/155/697067

In relation to IMAGE project and to geophysical interpretations above: "Electrical conductivity of Icelandic deep geothermal reservoirs up to supercritical conditions: Insight from laboratory experiments" (numerous tables and empirical laws for extrapolation)

https://www.sciencedirect.com/science/article/pii/S0377027317304092?casa_token=-tRHyFGSwmcAAAAA:Dmv27QlGdotqHm7Pp-GzsKgyoGSPmlFq70VKAq1w6rgWdT5n45q5xpBcy-OFh4eDYXksNHhwAf0

Also in relation to IMAGE project and to geophysical interpretations above:" The role of smectites in the electrical conductivity of active hydrothermal systems: electrical properties of core samples from Krafla volcano, Iceland"

https://academic.oup.com/gji/article-abstract/215/3/1558/5076040

b. Structure of the paper

The abstract could be shortened. I don't think the details on number and locations of samples are necessary, the two paragraphs l. 41 to l.53 could be significantly reduced.

The discussion is a bit overwhelming and seems to mix results and conclusions. It could be re-organized in different sub-sections, e.g.

(i) how is this database useful (see detailed questions suggestions above –> I think this section should be greatly enhanced and developed compared to how it is now)

(ii) what are the limitations and pitfalls and how to overcome them. More clarity in the discussion would help the reader feel more confident about in which contexts it is "safe" to use the database and in which contexts these data should be treated more carefully.

c. Materials and Methods - Electrical measurements

l. 442 "were executed in a similar way with an impedance spectrometer" –> you present three different types of electrical measurements, they are not that similar. Especially the "estimation from electrical resistivity tomographies performed in the sampling areas" l. 440. This is not state-of-the-art practice, so I would be careful here. How do you evaluate if the different types of measurements can be safely merged together? Have you tried different categories of measurements on the same samples? Alternatively, the different methods could be clearly presented in the database (different columns /

specific column for different methods). It is not clear to me where the "field samples" using ERT values are shown? Does this mean that people will be using ERT values to calibrate future MT inversion? Shouldn't that rather be handled by joint inversion? ERT has its own issues (equivalences, DC static shift, convergence of inversion) so the value of these data will strongly depend on how the ERT was carried out (electrode spacing, geometric factor, current injected, presence of background noise, misfit of the inversion). It can be a good idea to include electrical measurements from ERT in the database, especially if you have a lot of ERT surveys and few samples in the corresponding area, but they should be much more clearly explained. As a potential user of your database, I wouldn't use ERT values for calibration if I don't know how they have been obtained.

l. 449 "The error of measurements at dry conditions is 1.5% on average" –> how did you calculate it? It should be explained clearly. It is far from trivial to estimate this uncertainty. See examples below on the different sources of systematic errors and uncertainties in electrical measurements on rock samples.

http://eprints.whiterose.ac.uk/103298/

https://academic.oup.com/gji/article-abstract/215/3/1558/5076040

https://onlinelibrary.wiley.com/doi/full/10.1002/nsg.12069

c. Section 6 "Status of the database"

This section presents a lot of numbers, hard to follow, maybe a table would be better?

d. Discussion

l. 600 "The high number of analyzed plugs and samples enables detailed statistical and spatial geostatistical analyses on different 600 scales (plug, sample, outcrop, formation or model unit), spatial evaluation of the results in 2D or 3D or the validation of different analytical methods."

[Figure]

–> Electrical measurements were only made on 50 samples (Table 2), compared to 1000-1500 samples for all other properties. Is it sufficient for statistical analysis? Does it mean that this database has some specific limitations for interpreting MT/TEM inversions? If so, you should clearly state it and suggest how to overcome this issues (e.g. use data from IMAGE dataset or other studies mentioned above, where more than 100 different samples are presented per study, with all relevant mineralogical and petrophysical properties).

–> Why "only" 50 samples have electrical measurements? Some specific issues/limitations, maybe too time-consuming or expensive? I think it is totally normal to have limitations but it is important to the reader to understand the causes of this huge difference.

–> This is even more important given that some (how many??) of these 50 measurements are actually inferred from ERT and not direct laboratory measurements.

l. 608 "So far, only a few geothermal exploration studies in volcanic settings provide rock properties analyzed on [. . .] reservoir core samples".

–> There are more than few available. See references above and many other references. I think you should re-consider the structure and arguments of the discussion: see my other comments above. As I see it, the added value of your study is to provide a ready-to-use dataset for a specific exploration case + show and discuss how it can be used / not used in the future. Providing additional physico-chemical properties of volcanic rocks is of course a valuable side-effect. But it would not be sufficient as a single aim, because there are already a lot of data available, in particular in relation to IMAGE project.

l. 641 "In some cases, intensive hydrothermal alteration prevents a clear identification of the original rock type and correlation to equivalent units in the outcrops"

–> Good that you mention this limitation. What percentage of cases?

e. Figure 3 Electrical resistivity measurements are not part of the workflow figure. Why?

3. Technical corrections

- "Data" is a plural –> check throughout the manuscript

- "Aim at verb-ing" –> check

- Use of present/past –> try to choose one tense and keep it consistently. As it is now it makes the text hard to digest.

- Try to not overload sentences with adjectives, it makes it more difficult to read and slows down the flow. E.g. l. 611 "petrophysical and rock mechanical data was used for various different purposes."

Some sentences are difficult to understand, sometimes lacking a verb. Some examples below: - L. 644 "Current studies including detailed petrographic analyses and ICP-MS measurements, aiming to provide a better description and sample classification (Weydt et al., 2020, in prep.)." - L. 657 (which concept?) - L. 602 "Whenever possible, all parameters were analyzed on each plug allowing the identification of statistical and causal relationships between the parameters improving the accuracy of geostatistical predictions" –> this is hard to follow, maybe split the sentence?

Please also note the supplement to this comment:
https://essd.copernicus.org/preprints/essd-2020-139/essd-2020-139-RC2-supplement.pdf
* * *

---

## Author Comment (AC1) · 10 Oct 2020

Correspondence to: Leandra M. Weydt (weydt@geo.tu-darmstadt.de)

Author's comment on "Referee comment 1 – Review on Petrophysical and mechanical rock property database of the Los Humeros and Acoculco geothermal fields (Mexico)" by Anonymous Referee 1 We would like to thank the anonymous referee #1 (R1) for the helpful and valuable comments to improve our manuscript. In the following sections, we are addressing the referee's remarks and suggestions and present changes made in the manuscript.

Referee 1 – C2 line 68: "Another relevant, and recent, paper that the authors could consider citing here is Heap et al. (2020, JVGR). Heap, M. J., Gravley, D. M., Kennedy, B. M., Gilg, H. A., Bertolett, E., & Barker, S. L. (2020). Quantifying the role of hydrothermal alteration in creating geothermal and epithermal mineral resources: The Ohakuri ignimbrite (Taupo Volcanic Zone, New Zealand). Journal of Volcanology and Geothermal Research, 390,106703." Answer: Thank you very much for this very up to date reference. We added the citation "Heap et al. (2020)" to line 68.

Referee 1 – C2 lines 90, 107, 254: "Data is plural." Answer: Thank you very much for the detailed proofreading. The manuscript was checked again and the mistake was corrected accordingly.

Referee 1 – C2 line 146:" "Samples. . .were collected several times. . ." suggests that the same block of rock was collected several times. Suggest to reword." Answer: Thank you very much for this hint. We reworded the sentence as followed: "Whenever possible, each geological unit was sampled several times at different outcrop locations to cover the unit's heterogeneity and only samples with an overall fresh appearance unaffected by weathering were considered."

Referee1 – C2 line 265:" I suspect the authors mean "too friable" rather than "too brittle"." Answer: At this point, we wanted to describe samples that tend to break easily, which prohibited proper conduction of rock mechanical tests in terms of test duration and sample preparation. This accounts for limestones collected close to dykes and intrusive bodies containing several calcite-filled fractures, but also intensively hydrothermally altered lavas collected from a large fault zone located west of the Los Humeros Volcanic Complex. Thus, both words would be correct. We changed "brittle" to "fragile" to describe this phenomenon in a more generalized way.

Referee 1 – C2 line 329:" You mean "Table 2"?" Answer: Thank you very much for pointing this out. This sentence refers to the general description of the database presented in section 4 (lines 245 to 255) and not to Table 2 presented in the manuscript.

The database is provided in an Excel file containing two datasheets. The first datasheet contains all information on the analyzed petrophysical properties, while the second one includes all chemical data. To avoid any misunderstandings, we changed "the second table" to "the second datasheet of the database".

Referee 1 – C2 line 350:" A temperature of 105 _C might be high enough to encourage thermal microcracking or damage clays. Can the authors comment on the suitability of using this temperature? Are the authors sure the materials were not affected?" Answer: The samples were prepared according to internationally recognized standard methods (ASTM D4543, 2019, ASTM D4525, 2013), which recommend a temperature of approximately 100 °C for common rock samples. The majority of the samples contain no clays and samples affected by hydrothermal alteration such as the reservoir core samples contain mineral assemblages developed at much higher temperatures. The reservoir core samples were stored for more than twenty years at the CFE core storage. Chemical and petrographic analyses did not reveal any retrograde/low-temperature alteration products caused by weathering or humid storage conditions. Likewise, only outcrop samples with an overall fresh appearance were collected. A good indicator for mass losses and mineralogical changes are also the temperature and heat flow curves of the specific heat capacity measurements, whereby sample material was heated at a steady rate from 20 up to 200 °C. Furthermore, the sample weight was recorded before and after the measurements and significant mass losses due to the collapse of clay minerals were not observed. Thus, the effect of swelling clays or clays that are sensitive to temperature changes between 20 and 105 °C on the petrophysical rock properties can be neglected. In addition, the effect of thermal microcracking caused by thermal stress at temperatures between 20 and 100 °C can be neglected. Recent studies have shown that microfracture development and fracture opening start at 200 °C in basaltic lavas and tuff, at 500 °C in dolerite and gabbro (Siratovich et al., 2011), and at about 180°C in high-strength concrete (Heap et al., 2013). Even thermal stressing of up to 750 °C and subsequent cooling did not significantly impact the petrophysical and rock mechanical behavior of andesitic lavas (Heap et al., 2014, 2018),

while it had a more variable impact on tuff samples above 350 °C (Heap et al., 2012). Thereby, the thermal resilience of the samples can be explained by the thermal stability of the mineral assemblage (temperature-dependent break down of each mineral) and the presence of pre-existing microcracks (Heap et al., 2018). Generally, microcracks develop due to stress caused by mineral expansion and contraction during temperature increase and decrease, respectively. However, already existing microcracks close and reopen as a response to volumetric changes caused by temperature changes without further microcracking. Thus, samples that already underwent (thermal) fracturing and metamorphic processes do not tend to develop new microfractures, until they are exposed to higher temperatures than previously. This phenomenon is called the Kaiser temperature-memory effect and is presented in e.g. Vinciguerra et al. (2005), Heap et al. (2014, 2018), and recently in Vagnon et al. (2020 submitted), which also includes samples from this study (limestones from Las Minas). Especially, the metamorphic and hydrothermally altered rocks used in this study contain numerous fractures and microcracks and experienced temperatures much higher than 105 °C. Thus, the probability of rock property changes caused by the drying procedure exceeding the error of the measurement devices is very small.

Referee 1 – C2 line 353 and 439:" Can the authors comment on the effectiveness of saturating samples by leaving them submersed in water? For tight rocks, it seems doubtful that water would have penetrated thin pores/cracks. Errors resulting from incomplete saturation would influence, for example, the porosity measurements using the triple-weight method (Lines 365 and 370)." Answer: The majority of the porosity data provided in the database were performed using the combined helium and powder pycnometer method. The porosity measurements using the triple weighing and caliper method were performed additionally to the pycnometer method to compare and validate different analytical approaches. They were conducted according to internationally recognized testing methods (ISRM, 1981, and ISRM, 1979) and were repeated several times to provide a statistically verified mean value. However, the results of the different measurement techniques applied to the same sample material are well in line with

each other. Variations in particle density between different methods applied on the same samples range between 0.3-2.9% (coefficient of variation) for limestones with porosities smaller than 3 % and 0.5-3.5% for pyroclastic rocks with porosities between 11 and 15%. The variations in bulk density between the different methods are in the same range. Therefore, we can argue that the applied methods produce measurement results with sufficient precision.

Referee 1 – C2 line 360:" For those unsure of the meaning of "effective porosity", I would add "i.e. connected porosity" in parentheses here." Answer: Agreed. The sentence was changed to "Afterwards porosities were calculated from the resulting differences in volume and represent the gas-effective porosity, also known as connected porosity."

Referee 1 – C2 line 373:" What was the range of plug length? Measurements on "short" samples of a homogeneous sandstone were recently shown to provide reliable permeability values,see Heap (2019). These authors argued that permeability measurements on "short"samples are reliable as long as the pore/grain/crystal size is small compared to the length/diameter of the sample. Heap, M. J. (2019). The influence of sample geometry on the permeability of a porous sandstone. Geoscientific Instrumentation, Methods and Data Systems, 8(1), 55-61." Answer: The length of the plugs used for permeability measurements ranges between $\sim$ 20 mm and $\sim$ 80 mm. Thus, short and longer samples are considered in this study. We agree that measurements on small scale samples are reliable as long as the samples represent the minimal representative elementary volume (REV; Ringrose and Bentley, 2015) as described in section 4.1 line 316f.

Referee 1 – C3 line 377:" Gas permeability measurements for high-permeability samples and/or when using high flow rates likely also require a Forchheimer correction. Did the authors check for this?" Answer: Thank you very much for this question. The recorded flow rates of the gas permeability measurements performed at TUDA, GFZ and UNAM were checked for turbulent fluid flow (Forchheimer effect, Forchheimer,

1901). Therefore, the volumetric flow rates were plotted against the corresponding reciprocal permeability determined for each differential pressure. A Forchheimer correction is needed, whenever the data show a positive linear trend. This step needs to be performed before the Klinkenberg correction (gas slippage effect, Klinkenberg, 1941) to ensure that the recorded results meet the requirements of Darcy's law, which assumes laminar fluid flow (Kushnir et al., 2018, Heap, 2019). However, the recorded results obtained in this study didn't show a positive linear trend. Furthermore, we identified that permeability measurements corrected after Forchheimer were in most cases equal to the original values or at least in the same order of magnitude as the original values. Thus, a correction after Forchheimer was not necessary, because the corrected values are within the error range of the measurement device.

Referee 1 – C3 line 381:" ". . .at five pore fluid pressure levels. . ." " Answer: Agreed. We changed the sentence according to the reviewer's suggestion.

Referee 1 – C3 line 407:" Elastic wave velocities were measured parallel to the sample axis?" Answer: Thank you very much for that comment. The elastic wave velocities were measured along the sample axis. The transmitter-receiver transducers were pressed centrically against each parallel surface of the samples using a contact pressure of about 1 bar. To further clarify, we added "along the sample axis" in line 406 and the following sentence "Thereby, the transducers were pressed against the parallel surfaces of the samples using a contact pressure of about 1 bar." in line 411.

Referee 1 – C3 line 425:" Can the authors provide more information as to how the saturated velocities were measured? On samples submersed in water? Or were the samples wrapped in cling film and quickly measured to avoid desaturation?" Answer: Before measurements, the samples were stored in degassed and de-ionized water. After preparing the device, the samples were immediately installed between the transducers to perform the measurements. This procedure takes only a few seconds and the transmitted signals can be recorded a few times until the sample starts to desaturate, which of course affects the measurements. Wrapping up the samples in cling film

would not be practical for two reasons: 1) The cling film would not stick to the surface, because the sample is completely wet and wrapped in a water film, and 2) this step requires to take the sample out of the water before the measurement resulting in an earlier desaturation of the sample.

Referee 1 – C3 line 474:" If the authors prefer to use "G-Modulus", I would also put "shear modulus" in parentheses to avoid any confusion." Answer: Thank you very much for this comment. We added "also known as shear modulus" to line 473.

Referee 1 – C3 line 479:" This should be "load at failure/maximum load" and "cross-sectional area"." Answer: Thank you very much for this remark. We changed the sentence to "where F is the load at failure [N] and A is the cross-sectional area of the sample [mm$^2$]."

Referee 1 – C3 line 484:" Do the authors mean here that they used a constant loading rate of 0.5 kN/s? It's not clear. Written as it is, it suggests that the loading rate was variable and that the maximum was 0.5 kN/s." Answer: Thank you very much for pointing this out. At TU Darmstadt the destructive tests using the hydraulic uniaxial press were usually performed at 0.5 kN s-1. The exception form very soft or fragile samples, such as ignimbrites, pumice or intensively fractured limestones. For these samples, the load rate was individually reduced to 0.25 or 0.1 kN s-1 to meet the test requirements and to ensure the minimal test duration. Otherwise, the sample would break too quickly or immediately after starting the measurement resulting in invalid test results. We changed the passage accordingly.

Referee 1 – C3 line 485:" Do the authors mean the loading rate?" Answer: Yes, this is correct. We added "rate" to line 485 to avoid any misunderstandings.

Referee 1 – C3 line 490:" What type of sensor? Strain gauges?" Answer: For the determination of the axial displacement and lateral extension of the plugs during cyclic loading, LVDT sensors (linear variable differential transformer) were used. The detailed setup is described in DIN 18141-1:2014-05 and Mutschler (2004). To clarify, we added

"(LVDT sensors)" to line 490.

Referee 1 – C3 line 508:" Can the authors elaborate on what they mean by "tension controlled"?" Answer: Tension controlled means the loading of the sample is configured in MPa s-1. However, this sentence needs to be corrected. The samples at TU Delft were also tested 'force-controlled' at 0.15 kN s-1.

Referee 1 – C3 line 519:" They were loaded diametrically in compression?" Answer: Yes, this is correct. The samples were loaded diametrically in compression. The detailed test setup is described in ASTM 3967 (2016) and Lepique (2008). To avoid misinterpretations, we added "also known as diametrical compression" to line 519.

Referee 1 – C3 line 537:" The triaxial experiments were performed on dry samples?" Answer: This is correct. The triaxial testing device at TU Darmstadt is equipped to perform measurements on dry samples only. Unless otherwise stated, all measurements are performed on oven-dry samples (see Figure 3 in the manuscript). Only thermal conductivity, thermal diffusivity, P-wave and S-wave velocity, as well as electric resistivity, were analyzed at dry and saturated conditions. To point this out, "oven-dry" was added to line 539.

Referee 1 – C3 line 607:" See also the study by Eggertsson et al. (2020), who measured samples taken from the Krafla geothermal system in Iceland. Eggertsson, G. H., Lavallée, Y., Kendrick, J. E., & Markússon, S. H. (2020). Improving fluid flow in geothermal reservoirs by thermal and mechanical stimulation: The case of Krafla volcano, Iceland. Journal of Volcanology and Geothermal Research, 391, 106351." Answer: Thank you very much for referring to this article. We added the quotation Eggertsson et al. (2020) to line 608.

Referee 1 – C4 line 669:" I think the authors should include an additional paragraph(s) that states that large-scale modelling, such as fluid circulation models, require up-scaled values not those measured in the laboratory. I think it would be beneficial for the reader if the authors explain the issues surrounding using laboratory-measured

values in large-scale models and discuss/present existing methods typically used to upscale such values." Answer: We agree that a comment on this topic would be very beneficial for the reader. However, upscaling of reservoir data and problems that occur using laboratory measurements in large-scale models is a broard topic and could easily fill a whole new paper, which is beyond the scope of this article. The applied upscaling methods and problems that might occur, strongly depend on the purpose, size, and accuracy of the model. As it is not possible to go into much detail in this manuscript, we added a short paragraph to section 7 Discussion to shortly discuss the required steps before using the data in reservoir models.

Further modifications: The numbers included in Table 2 in section 6 "Status of the database" were adjusted as further measurement results were added to the database.

  References

ASTM D3967-16, Standard Test Method for Splitting Tensile Strength of Intact Rock Core Specimens, ASTM International, West Conshohocken, PA, USA, 5 pp., DOI:10.1520/D3967-16, 2016.

ASTM D4525-13e2: Standard Test Method for Permeability of Rocks by Flowing Air, ASTM International, West Conshohocken, PA, USA, 5 pp., DOI: 10.1520/D4525-13E02, 2013.

ASTM D4543-19: Standard Practices for Preparing Rock Core Specimens and Determining Dimensional and Shape Tolerances, ASTM International, West Conshohocken, PA, USA, 13 pp., DOI: 10.1520/D4543-19, 2019.

Bär, K., Arndt, D., Fritsche, J.-G., Kracht, M., Hoppe, A.,and Sass, I.: 3D-Modellierung der tiefengeothermischen Poten-ziale von Hessen – Eingangsdaten und Potenzialausweisung, Z.dt. Ges. Geowiss., 162, 371–388, 2011.

DIN 18141-1:2014-05: Baugrund - Untersuchung von Gesteinsproben - Teil 1: Bestimmung der einaxialen Druckfestigkeit, Beuth, 14 pp.,

https://dx.doi.org/10.31030/2100323, 2014.

Eggertsson, G. H., Lavallée, Y., Kendrick, J. E., and Markússon, S. H.: Improving fluid flow in geothermal reservoirs by thermal and mechanical stimulation: The case of Krafla volcano, Iceland. Journal of Volcanology and Geothermal Research, 391, 106351, 2020.

Forchheimer, P.:Wasserbewegung durch Boden, Z. Ver. Dtsch. Ing., 45, 1782–1788, 1901. Heap, M. J.: The influence of sample geometry on the permeability of a porous sandstone. Geoscientific Instrumentation, Methods and Data Systems, 8(1), 55-61, 2019.

Heap, M.J., Lavallée, Y., Laumann, A., Hess, K.U., Meredith, P.G., Dingwell, D.B.: How tough is tuff in the event offire? Geology 40 (4), 311–314, 2012.

Heap, M. J., Lavallée, Y., Laumann, A., Hess, K.-U., Meredith, P. G., Dingwell, D. B., Huismann, S., and Weise, F.: The influence of thermal-stressing (up to 1000°C) on the physical, mechanical,and chemical properties of siliceous-aggregate, high-strength concrete, Construction and Building Materials, 42, 248-265, 2013.

Heap, M.J., Lavallée, Y., Petrakova, L., Baud, P., Reuschlé, T., Varley, N.R., Dingwell, D.B.: Microstructural controls on the physical and mechanical properties ofedifice-forming andesites at Volcán de Colima, Mexico. J. Geophys. Res. Solid Earth119 (4), 2925–2963, 2014.

Heap, M. J., Coats, R., Chen, C., Varley, N., Lavallée, Y., Kendrick, J., Xu, T., Reuschlé, T.: Thermal resilience of microcracked andesitic dome rocks. Journal of Volcanology and Geothermal Research, 367, 20-30, doi:10.1016/j.jvolgeores.2018.10.021, 2018.

ISRM: Suggested methods for determining the uniaxial compressive strength and deformability of rock materials, International Society for Rock Mechanics, Pergamon Press, 137 - 138, 1979.

ISRM: Suggested Methods – Rock characterization testing and monitoring, Ed. E.T.

Brown, published for the commission on testing methods, International Society for Rock Mechanics, Pergamon Press, 210 pp., 1981.

Klinkenberg, L. J.: The permeability of porous media to liquids and gases, Drilling Production Practice, API, 200–213, 1941.

Kushnir, A. R. L., Heap, M. J., and Baud, P.: Assessing the role of fractures on the permeability of the Permo-Triassic sandstones at the Soultz-sous-Forêts (France) geothermal site, Geothermics, 74, 181–189, https://doi.org/10.1016/j.geothermics.2018.03.009, 2018.

Lepique, M.: Empfehlung Nr. 10 des Arbeitskreises 3.3 "Versuchstechnik Fels" der Deutschen Gesellschaft für Geotechnik e. V.: Indirekter Zugversuch an Gesteinsproben – Spaltzugversuch, Bautechnik, 85, 623–627, DOI: 10.1002/bate.200810048, 2008.

Mutschler, T.: Neufassung der Empfehlung Nr. 1 des Arbeitskreises "Versuchstechnik Fels" der Deutschen Gesellschaft für Geotechnik e. V.: Einaxiale Druckversuche an zylindrischen Gesteinsprüfkörpern, Bautechnik, 81, 825-834, doi: 10.1002/bate.200490194, 2004. Ringrose, P., and Bentley, M.: Reservoir Model Design, Springer Netherlands, 249 pp., http://dx.doi.org/10.1007/978-94-007-5497-3, 2015.

Siratovich, P. A., Sass, I., Homuth, S., and Bjornsson, A.: Thermal Stimultaion of Geothermal Reservoirs and Laboratory Investigation of Thermally Induced Fractures, GRC Transactions, 35, 1529-1536, 2011.

Vagnon; F., Colombero, C., Comina, C., Ferrero, A. M., Mandrone, G., Missagia, R., Vinciguerra, S. C.: Temperature effects on physical properties of carbonate rocks, Rock Mechanics and Geotechnical Engineering, 2020 in review.

Vinciguerra, S., Trovato, C., Meredith, P.G., Benson, P.M.: Relating seismic velocities, thermal cracking and permeability in Mt. Etna and Iceland basalts. Int. J. Rock

Mech.Min. Sci. 42 (7–8), 900–910, 2005.

Please also note the supplement to this comment:
https://essd.copernicus.org/preprints/essd-2020-139/essd-2020-139-AC1-supplement.pdf

---

## Author Comment (AC2) · 10 Oct 2020

Correspondence to: Leandra M. Weydt (weydt@geo.tu-darmstadt.de)

Author's comment on "Referee comment 2 – Review on Petrophysical and mechanical rock property database of the Los Humeros and Acoculco geothermal fields (Mexico)" by Léa Lévy

We would like to thank Léa Lévy as referee #2 (R2) for her valuable comments and suggestions to improve our manuscript. In the following sections, we would like to address the two main remarks of R2 regarding 1) "the lack of electrical measurements

and the implications for statistical analyses and MT/TEM/DC survey interpretations" as well as 2) referencing to similar studies and a more detailed explanation of the limitations of this specific database. More detailed questions regarding specific text sections are listed below and we hope that our reworked manuscript provides the requested clarification.

General remarks: Referee 2 – Remark 1: "More explanations about the lack of electrical measurements (50 samples versus 1000-1500 for all other properties) and the implications for statistical analyses and MT/TEM/DC surveys interpretations. MT and TEM are among the most common methods (if not the most) used in geothermal exploration, so this discussion is critical to justify the usefulness of your paper. The paper also should help the reader find ways to overcome this gap. Additionally, the use of ERT for inferring the resistivity of samples is not a state-of-the-art method and it is not clear how many of the 50 samples are inferred from ERT. Especially because 50 samples have a formation factor, which I guess you cannot obtain with only ERT measurements. More clarity is needed here." Answer: The database presented in our manuscript is the result of a joint effort of multiple project partners of the GEMex project working on the task of petrophysical rock characterization. We joined forces so that each partner involved in this task was performing the measurements which were available at their institutions and which were part of their main expertise. Given the different amount of person-months allocated to the individual partners for each task, different numbers of measurements could be performed by the different partners. In the particular case of the electric resistivity measurements, the availability of measurement devices and logistical problems in the project were the main reason for the comparatively "low" number of these measurements. Additionally, the main purpose of this database was not to provide input data for geophysical exploration methods such as MT/TEM/DC and their interpretation but to provide input data for numerical reservoir models mainly focussing on their thermo-hydro-mechanical behaviour during exploitation. Regarding electric resistivity measurements, the 50 existing measurements with a formation factor in the database were analyzed on plugs in the laboratory. Field measurements were performed on 24 samples, which are marked as 'field samples' in the database. However, we recognized that the electric resistivity measurements of these samples were lacking in the database (so far only the P-wave measurements were included), which might be the reason for the referee's questions. To accomplish the reviewer observations and given the underlined uncertainty in field ERT data, the electric resistivity data inferred from ERT and the corresponding text passage will be removed from the database and the manuscript. Hence, about 15% of the outcrop samples included in the database were analyzed for electric resistivity covering several lithologies from the basement to the caprock. We think that this amount of data is already supporting MT/TEM/DC survey interpretations. However, we agree, that it would have been very useful for the statistical analysis of electrical properties of the investigated samples and their usefulness for MT/TEM/DC survey interpretation if we would have done a more comprehensive measurement programme for these properties as well. But as also stated in our manuscript, the reservoir properties are strongly governed by the tectonic overprint and the resulting faults, damage zones and fracture networks locally increasing the porosity and permeability of the geological succession and as such act as conduits for reservoir fluids, which would also be the strongest detectable anomalies in electromagnetic surveys. See also further comments below.

Referee 2 – Remark 2: "(ii) Clearer aim and context. References and comparison to recent, similar and complementary studies are lacking. Especially to feed the discussion on how to overcome limitations of this specific database. I have suggested a few studies that I know of and consider complementary." Answer: The arguments of referee 2 strongly focus on electric resistivity measurements and MT/TEM/DC survey interpretation. As mentioned above, this was not the main purpose of the GEMex project and this study. Referee 2 considered several interesting articles about Iceland, which predominantly cover the interpretation of electrical resistivity tomography (ERT), gravimetric and seismic surveys (sometimes in context with porosity, density or permeability data), and some detailed analyses of electric resistivity and magnetic susceptibility measurements on a small number of rock samples regarding super-critical conditions or hydrothermal alteration. Since the GEMex project focuses on deep super-hot geothermal systems, findings and raw data from the Krafla geothermal field are indeed good complementation. However, the suggested articles barely contain rock properties or only focus on detailed analyses of one single parameter. Thus, they do not represent a "database" as defined in our study, in which a high number of samples were analyzed for a wide variety of parameters. As our study focuses on rock properties associated with volcanic settings and/or super-hot geothermal systems, we already included several studies in our manuscript presenting newly generated rock property data from different study areas that fit the context of our work, but also studies that represent petrophysical rock characterization and reservoir characterization in a wider context. Introduction: General literature: 1. Schön (2015) = general introduction into rock properties; 2. Bär et al. (2020) = recently published Image database containing digitized rock property data from published articles; 3. Clauser and Huenges (1995) = average thermal conductivity values for minerals and specific rock types, 4. Sass and Götz (2012) = thermofacies concept considering thermal conductivity and permeability data for reservoir characterization; 5. Howell et al. (2014) = the application of outcrop analogues in geomodelling, upscaling; 6. Linsel et al. (2020) = chemical and petrophysical characteristics of sandstone on the lithofacies scale, Germany; Hydrothermal alteration: 7. Mielke et al. (2015) = thermophysical properties, Tauharo geothermal fiel, New Zealand; 8. Pola et al. (2012) = petrophysical and rock mechanical properties, Solfatara crater, Ischia Island and Bolsena volcanic zone, Italy; 9. Mordensky et al. (2019) = petrophysical and rock mechanical properties, Mt. Ruapehu, New Zealand; 10. Durán et al. (2019) = petrophysical properties, Ngatamariki geothermal field, New Zealand; Diagenetic processes: 11. Aretz et al. (2015) = petrophysical properties related to mineral content, depositional environment and diagenesis, sandstone, Upper Rhine Graben, Germany; 12. Weydt et al. (2018a) = petro- and thermophysical properties related to dolomitization, Devonian aquifer systems, Alberta; Literature related to the study area in Mexico: 13. Weydt et al. (2018b) = primilary results of the GEMex project; 14. Contreras et al. (1990) = petrophysical and rock mechanical

properties, reservoir core samples, Los Humeros; 15. García-Gutiérrez and Contreras (2007) = thermal conductivity measruements, reservoir core samples; Los Humeros; 16. Canet et al. (2015) = petrophysical properties, reservoir core samples, Acoculco. Project framework: 17. López-Hernández et al. (2009) = exploration study regarding hydrothermal alteration, Acoculco; 18. Lepillier et al. (2019) = rock mechanical properties obtained on samples from Las Minas used for DFM modeling; 19. Kummerow et al. (2020) = electrical and hydraulic properties at supercritical conditions; 20. Deb et al. (2019c) = Laboratory fracturing experiments on big blocks from Las Minas; 21. Lacinska et al. (2020) = fluid-rock reactions analyzed on outcrop and reservoir core samples from Los Humeros. Discussion: 22. Lenhardt and Götz (2011) = petro- and thermophysical properties of volcanic rocks, Central Mexico; 23. Pola (2014) = rock mechanical properties , Solfatara crater, Ischia Island and Bolsena volcanic zone, Italy; 24. Mielke et al. (2016) = petro- and thermophysical properties, Taupo Volcanic Zone, New Zealand; 25. Heap and Kennedy (2016) = scale-dependent permeability of andeistic lavas, Mt. Ruapehu, New Zealand; 26. Navelot et al. (2018) = petrophysical, thermophysical, dynamic mechanical properties of various volcanic rocks and the impact of hydrothermal alteration, Guadeloupe Archipelago, West Indies, Antilles; 27. Stimac et al. (2004) = petrophysical properties of andesitic lavas, Tiwi geothermal field, Philippines; 28. Siratovich et al. (2014) = petrophysical and mechanical properties of andesitic lavas, Rotokawa geothermal field, New Zealand; 29. Ólavsdóttir et al. (2015) = reservoir quality of volcaniclastic units, Faroe Islands, northeast Atlantic; 30. Cant et al. (2018) = permeability of different volcanic rocks, Ngatamariki geothermal field, New Zealand. Since most of these studies are published later than 2010, we don't see a lack of 'recent and similar' studies cited and discussed in our manuscript. However, we agree that literature from Iceland would be good complementation and we added some of the below-mentioned studies to the discussion. Furthermore, we agree that a short section regarding other extensive databases (like oil and gas databases, petrological or rock chemical databases) could be very beneficial for the reader and we added a short section to the manuscript. Further comments are included in the sections below.

Specific comments: Comments on additional references: Referee 2: "Is this database only intended at interpreting geophysical datasets in the two corresponding areas in Mexico, for this corresponding deepEGS exploration project? Or do you see possibilities to use this database at other geothermal fields, for different geothermal exploration projects? → If this is only for the present exploration project in Mexico, are you then suggesting that such extensive data collection be done for every geothermal system to be explored? It would be interesting to get an idea of how much resource it requires, compared to other exploration costs. Is it realistic? Are we going to need public funds for every new geothermal exploration project? Or is there a point where we will have hopefully collected enough petrophysical data and run sufficient statistical analyses, to be able to build experience from one field to the other, and even compare fields world-wide? → If you consider that this database can be used in other contexts than in Mexico, it would be very valuable to elaborate a bit. How can a given petrophysical dataset be used to better understand reservoir behavior? To interpret geophysical data at other places¿' Answer: The main purpose of the GEMex project is to develop new and transferable approaches for the exploration and development of super-hot and unconventional geothermal systems worldwide. Thus, this database not only intends to provide comprehensive and detailed input data for numerical modelling and to support the interpretation of geophysical surveys performed in Acoculco and Los Humeros, but also to serve as an example for other and future geothermal exploration studies. The accuracy of 3D geological models strongly depends on the amount and quality of input data, which are often lacking especially in early exploration stages. Depending on the scale of the model (global or regional) or for a first assessment, it is often sufficient to use literature data from a similar geological context (for example using this data for other volcanic settings in the TMVB or elsewhere), also from an economical point of view. However, since our data and data from literature have shown, that rock properties strongly depend on the original texture, mineral composition, pore and fracture distribution of the rocks as well as tectonic, diagenetic and metamorphic processes resulting in a high geological variability, it is always favorable or necessary to investigate the

relevant key formations in a study area – especially for small-scaled investigations it is deemed necessary. Outcrop analogue studies represent a cost-effective approach to investigate the different geological formations in the study area and should be included in exploration programs, especially in greenfields, where the overall knowledge of the geological setting is still low. Of course, the extent of such studies and the amount of data that need to be collected strongly depends on the size of the study area and the purpose of the project. However, when generating new data, it is necessary that results from different institutes or disciplines are uniform and can be correlated with each other. Therefore, the coordination of field work and laboratory measurements to combine different disciplines, as it has been performed in this study, enabled the compilation of this large amount of data and also reduced costs in the field and for the shipment. In addition, the compilation of such datasets and the creation of databases also always represents a learning curve. Since super-hot or supercritical geothermal systems are a relatively new topic and operating and drilling into these systems is very challenging, we are still at a point, at which we need to better understand the processes triggering these systems. Finding similarities between several systems, provides the possibility to transfer knowledge, exploration and exploitation approaches/technologies. With respect to our study, e.g. the andesitic lavas of the geothermal reservoir in Los Humeros seem to be very similar to the Rotokawa andesitic lavas of the Rotokawa geothermal field in New Zealand (Siratovich et al., 2014) regarding the type and degree of hydrothermal alteration and rock properties. Furthermore, thermal weakening of carbonatic basements as described in Heap et al. (2013), was also observed in the few available reservoir core samples from Los Humeros and Acoculco. A more general finding is, that these systems are predominantly fracture controlled. These information are very valuable when it come to modelling. Therefore, it is indeed the objective of such databases, as the one suggested by us, to compile a large amount of data to at one point be able to draw conclusions also for other fields based on sufficient statistical evaluation.

Referee 2: "Regardless the answer to the question above, I think it is necessary to

Interactive
comment

put this study more in context with similar studies. It is not the first time that such a massive effort is made in the context of high-enthalpy geothermal exploration. I can see that you refer to the P3 database made in the frame of IMAGE project, where the focus was on Iceland and Italy. I think it is critical to expand a bit on the differences and on the coherence between the two projects / databases. Why is this new database necessary after the one in the IMAGE project? How are they complementary? What results from IMAGE have convinced you that making such a database was useful? This would be a useful addition at lines 80-87." Answer: We agree that it would be beneficial for the reader to point out to similar extensive databases and we will add a short section to the manuscript. The IMAGE database (Bär et al., 2020) collected, digitized and organized rock property data of 316 research articles and student theses including 75.573 data points of 28 different rock properties analyzed on a wide variety of lithologies worldwide. While the IMAGE database is an important resource to enhance future modelling approaches and significantly increased the availability of standardized rock properties, it only contains very limited number of data points or parameters for each formation or area investigated. Furthermore, a detailed sample description is not always available. The usage of data from the IMAGE database was not sufficient for the purpose of the GEMex project regarding the level of detail and the geological complexity in the study area. In contrast, the database presented in this manuscript, contains more than 31.000 data points and 34 different parameters for one study area only. The main difference to the IMAGE database is, that all samples were analyzed the same way. Whenever possible, each parameter was analyzed on each plug. This approach significantly simplifies and improves statistical analyses and allows for correlation between different parameters. While recent research articles usually focus on one single target formation or a specific rock type, this database covers all relevant geological formations from the basement to the caprock covering a wide range of sedimentary, volcanic, igneous and metamorphic rocks. Also the sampling strategy (collecting samples from the same formation from different outcrop locations within the study area) lead to an improved understanding of the spatial variability of each individual unit. The

amount of data and the level of detail presented in this study significantly improved the geological understanding of the study area, but also helps to better understand the relation between different rock parameters and how they are affected by different processes (e.g. fracturing or hydrothermal alteration). This is useful to derive general trends (also in combination with other data, IMAGE) for e.g. numerical modelling or to go one step further and use such data to train machine learning algorithms for rock property prediction.

Referee 2: Suggestions of additional references "Imaging the magmatic system beneath the Krafla geothermal field, Iceland: A new 3-D electrical resistivity model from inversion of magnetotelluric data" → Interpret MT inversions at geothermal fields using petrophysical calibration (especially temperature dependent measurements).

"New Conceptual Model for the Magma-Hydrothermal-Tectonic System of Krafla, NE Iceland" https://www.mdpi.com/2076-3263/10/1/34 → Shows how conceptual models are regularly updated in light of new petrophysical understandings

Study related to both IMAGE and GEMEx projects: "Electrical resistivity tomography and time-domain induced polarization field investigations of geothermal areas at Krafla, Iceland: comparison to borehole and laboratory frequency-domain electrical observations" https://academic.oup.com/gji/article-abstract/218/3/1469/5497301 → Interpret DC/IP inversions based on petrophysical measurements on core samples at the exact same site. Discussion on upscaling with in particular comparison of samples to borehole logging and analyses of in-situ versus laboratory temperature differences.

"A probabilistic geologic model of the Krafla geothermal system constrained by gravimetric data" https://geothermal-energy-journal.springeropen.com/articles/10.1186/s40517-019-0143-6 → Statistical analysis of the link lithology versus density, and use to interpret gravity data. Could be cited around l. 79.

"Subsurface imaging of water electrical conductivity, hydraulic permeability and lithology at contaminated sites by induced polarization" https://academic.oup.com/gji/article/213/2/770/4816733 → Lithology and permeability characterization in Denmark using petrophysical calibration based on extensive laboratory database (sedimentary context) Answer: This study focuses on contaminated groundwater of a very shallow sand and clayrich aquifer in Denkmark. Our study focuses on deep high-enthalpy geothermal reservoirs and rock properties. We don't see any connection between these two studies.

You are saying in the introduction that data are distributed in different places (l.72-79), which makes their use complicated. But if this database is intended to be used in a more general manner than just in this project in Mexico, then there needs to be a (short) section on other similar database and how they can be combined. It could be in the discussion as well. I would also add references, either in the introduction (near l.72-79) or in the discussion, to data collection presented in separated papers or PhD thesis, provided that the data collection is significantly large and well-presented and contains consistent data to be comparable to your database, of course. That way the reader will know where to find complementary information, if he needs, e.g. in the IMAGE database or in other articles. A few suggestions below.

"Modification of the magnetic mineralogy in basalts due to fluid–rock interactions in a high-temperature geothermal system (Krafla, Iceland)" (see Table A1) https://academic.oup.com/gji/article/186/1/155/697067

In relation to IMAGE project and to geophysical interpretations above: "Electrical conductivity of Icelandic deep geothermal reservoirs up to supercritical conditions: Insight from laboratory experiments" (numerous tables and empirical laws for extrapolation)

Also in relation to IMAGE project and to geophysical interpretations above:" The role of smectites in the electrical conductivity of active hydrothermal systems: electrical properties of core samples from Krafla volcano, Iceland" https://academic.oup.com/gji/article-abstract/215/3/1558/5076040

Subsurface imaging of water electrical conductivity, hydraulic permeability and lithology at contaminated sites by induced polarization -> kontaminiertes Grundwasser in Dänemark, sand and clays Answer: As mentioned above, we agree that it would be beneficial for the reader to point out further extensive databases. Some of the suggested research articles represent good examples of how rock property data can be used to interpret geophysical surveys and we will add some of the references to the discussion.

Comments on the structure of the database: Referee 2 – line 1: "The abstract could be shortened. I don't think the details on number and locations of samples are necessary, the two paragraphs l. 41 to l.53 could be significantly reduced." Answer: We agree that the abstract should present the content of the paper in an informative but concise manner. However, the paragraphs from lines 41 to 53 shortly describe the aim of the study, where and how the samples were collected and analyzed. This information is essential for the reader to quickly understand the geological context and to estimate whether the data might be useful for their interests or not. Furthermore, the listing of the analyzed parameters is of great importance for people actually working on databases or searching for specific data. Since the length of the abstract is far below 500 words and this paper represents a database in a data journal, we see no need for further changes here.

Referee 2 – line 595: "The discussion is a bit overwhelming and seems to mix results and conclusions. It could be re-organized in different sub-sections, e.g. o (i) how is this database useful (see detailed questions suggestions above → I think this section should be greatly enhanced and developed compared to how it is now) o (ii) what are the limitations and pitfalls and how to overcome them. More clarity in the discussion would help the reader feel more confident about in which contexts it is "safe" to use the database and in which contexts these data should be treated more carefully. " Answer: The discussion already describes why the data is useful and mentions several examples of how the data was used and will be used within the scope of the GEMex

project or can be used for other applications. We also discussed several limitations regarding the modeling of the Los Humeros and Acoculco caldera complexes or other future applications. Since this database allows for a wide range of future applications, it is not possible and also not the aim of this database to discuss all possible limitations that might come along with using this data. The samples and applied methods are well described and future users need to verify themselves whether this data are useful for their purposes or not. Therefore, we see no need in completely re-organizing this section. However, we agree that it would be beneficial for the reader to add more examples of how rock properties can be used in exploration studies and we added a short section to the first part of the discussion. We also improved the discussion regarding the varying number of measurements per analyzed parameter.

Comments on "Material and methods": Referee 2 – line 442: "were executed in a similar way with an impedance spectrometer" → you present three different types of electrical measurements, they are not that similar. Especially the "estimation from electrical resistivity tomographies performed in the sampling areas" l. 440. This is not state-of-the-art practice, so I would be careful here. How do you evaluate if the different types of measurements can be safely merged together? Have you tried different categories of measurements on the same samples? Alternatively, the different methods could be clearly presented in the database (different columns / specific column for different methods). It is not clear to me where the "field samples" using ERT values are shown? Does this mean that people will be using ERT values to calibrate future MT inversion? Shouldn't that rather be handled by joint inversion? ERT has its own issues (equivalences, DC static shift, convergence of inversion) so the value of these data will strongly depend on how the ERT was carried out (electrode spacing, geometric factor, current injected, presence of background noise, misfit of the inversion). It can be a good idea to include electrical measurements from ERT in the database, especially if you have a lot of ERT surveys and few samples in the corresponding area, but they should be much more clearly explained. As a potential user of your database, I wouldn't use ERT values for calibration if I don't know how they have been obtained."

Answer: In both laboratories, the electrical resistivities were measured with 4-electrode layouts. Although at GFZ sample resistivities were gained from measurements with an impedance spectrometer, SIP data are not part of the publication, and given resistivities are related to a fixed frequency of 1 kHz. Nonetheless, we recognized an inconsistency in the database, as resistivities were measured at UNITO for saturation with a 1000 $\mu$S/cm-fluid, while for GFZ those resistivities were given for saturation with a 10 mS/cm-fluid. To make the dataset comparable we changed the database and now all resistivities at saturated conditions are related to measurements at 1 mS/cm fluid conductivity. Unfortunately, it was not possible to perform a statistically verified evaluation of the different measurement methods and there is only a small overlap of the sample sets sent to GFZ and UNITO (about 3 samples). However, both institutes mainly analyzed the same lithologies and the individual samples were collected from the same outcrops or at least the same sampling area. Hence, we think that the measurements of both institutes are comparable. We completely agree with referee 2 with respect to the uncertainties and limitations of ERT. As explained above, the electric resistivity inferred from ERT data were accidently missing in the database. However, we decided to follow the arguments of referee 2 and will not include field measurements. Adding further columns with respect to the adopted methodologies will in our opinion reduce its visibility. As explained in the manuscript, the column 'Institution' in the databse is used to relate to the applied measurement methods. We, therefore, would prefer to not include further specifications here. We are available to perform this if the reviewer deems this mandatory for publication.

Referee 2 – line 449: "The error of measurements at dry conditions is 1.5% on average" → how did you calculate it? It should be explained clearly. It is far from trivial to estimate this uncertainty. See examples below on the different sources of systematic errors and uncertainties in electrical measurements on rock samples. Answer: We agree with the reviewer and have modified the corresponding text passage.

Comment on "Status of the database": Referee 2 – line 563: "This section presents a

lot of numbers, hard to follow, maybe a table would be better?" Answer: This section was intended to present an overview of the amount of data presented in the database regarding the study area, model units, and analyzed parameters. Therefore, one table (Table 2) and two figures (Fig. 6 and Fig. 7) were included. Since the two paragraphs in this section are relatively short, we see no need to add another table considering the already critical length of the manuscript.

Comments on "Discussion": Referee 2 – line 600: "The high number of analyzed plugs and samples enables detailed statistical and spatial geostatistical analyses on different 600 scales (plug, sample, outcrop, formation or model unit), spatial evaluation of the results in 2D or 3D or the validation of different analytical methods." –> Electrical measurements were only made on 50 samples (Table 2), compared to 1000-1500 samples for all other properties. Is it sufficient for statistical analysis? Does it mean that this database has some specific limitations for interpreting MT/TEM inversions?If so, you should clearly state it and suggest how to overcome this issues (e.g. use data from IMAGE dataset or other studies mentioned above, where more than 100 different samples are presented per study, with all relevant mineralogical and petrophysical properties). –> Why "only" 50 samples have electrical measurements? Some specific issues/limitations, maybe too time-consuming or expensive? I think it is totally normal to have limitations but it is important to the reader to understand the causes of this huge difference. –> This is even more important given that some (how many??) of these 50 measurements are actually inferred from ERT and not direct laboratory measurements. Answer: As explained above, all 50 electric resistivity measurements included in the database were obtained in the laboratory. The differences in the number of analyzed samples were caused by the availability of measurement devices and logistical problems within the project (e.g. shipment of equipment for the fieldwork, delay in the shipment of the samples back to Europe). Only two laboratories were equipped with appropriate measuring devices. Moreover, a limited amount of sample material had to be distributed between the partners. To summarize, the limitation of resistivity data resulted from a combination of logistical issues and time-consuming
measurements. The same accounts for specific heat capacity, fracture-toughness and triaxial measurements, which require a specific sample size and/or are relatively time-consuming and for which only one appropriate device was available in the consortium. For example, to obtain cohesion and friction angle, a minimum of three large plugs with a diameter of 55 mm and a length of 110 mm are required. This is a lot of sample material considering the sample preparation procedure (it is not easy to drill such large plugs as most of the samples contained a high number of fractures) and the extra effort to obtain such large boulders in the field (very limited access to the outcrops and requires a lot of equipment). Likewise, one single specific heat capacity measurement takes 24 hours. Since more than 200 samples were analyzed for this study in the end, it requires more than a year to obtain this amount of data. As a consequence, this parameter was obtained only once per sample, while other parameters were analyzed on each plug. This means that the total number of measurements per parameter given in Table 2 is not a criterion for "high or low number of measurements". As mentioned above, about 15% of all outcrop samples included in the database were analyzed for electric resistivity measurements. This amount of data usually fills a common research article and we think that it already supports the interpretation of TEM surveys as it covers all relevant lithologies in the study area. Therefore, we don't see a critical issue here as claimed by referee 2, although we agree that further data would be beneficial to improve statistical evaluation. We added a corresponding statement to the discussion.

Referee 2 – line 608: "So far, only a few geothermal exploration studies in volcanic settings provide rock properties analyzed on [. . .] reservoir core samples". –> There are more than few available. See references above and many other references. I think you should re-consider the structure and arguments of the discussion: see my other comments above. As I see it, the added value of your study is to provide a ready-to-use dataset for a specific exploration case + show and discuss how it can be used / not used in the future. Providing additional physico-chemical properties of volcanic rocks is of course a valuable side-effect. But it would not be sufficient as a single aim, because there are already a lot of data available, in particular in relation to IMAGE

project." Answer: We agree that this sentence can be interpreted in different ways and should be specified. Compared to siliciclastic or carbonate basins used for oil and gas exploration, the amount of petrophysical and mechanical rock property data for volcanic settings in the context of high-enthalpy geothermal systems is less documented. Furthermore, the effect of hydrothermal alteration and metamorphic processes on the rock properties is only addressed in a few studies so far (Pola et al., 2012, Frolova et al., 2014, Mielke et al., 2015, Navelot et al., 2018, Mordensky et al., 2019, Heap et al., 2019, Delayre et al., 2020) and is not fully understood yet. Up to now, there exist only very few studies, that actually compare rock properties obtained on reservoir core samples with stratigraphically equivalent formations in outcrops. The increased interest in super-hot or supercritical geothermal reservoirs for electricity generation also increased the demand for raw data for numerical modeling and the interpretation of geophysical exploration surveys. Thus a profound understanding of rock properties and how they are affected is essential. As there were no such data available for the GEMex project, this study was initiated to overcome this knowledge gap and to avoid using generalized data from the literature. See also the comments above.

Referee 2 – line 641: ""In some cases, intensive hydrothermal alteration prevents a clear identification of the original rock type and correlation to equivalent units in the outcrops" –> Good that you mention this limitation. What percentage of cases?" Answer: The identification of about 25% of the reservoir core samples were problematic. These samples are marked as "undefined altered lava" in the database. We added this information to line 641.

Comment on figures: Referee 2 – Figure 3: "Electrical resistivity measurements are not part of the workflow figure. Why¿' Answer: Figure 3 represents the schematic workflow of the sample preparation and measurement procedure at TU Darmstadt. Unfortunately, it was not possible to perform electricity measurements at this stage of the project at TU Darmstadt. As described in section 5 "Material and methods" these measurements were performed on selected sample material at GFZ and UNITO.

As the majority of the samples and parameters were analyzed at TU Darmstadt, it seemed plausible to us to illustrate the measurement procedure from this institute to demonstrate the general workflow in the laboratory.

Technical corrections: Comments on technical corrections were carefully read and considered during the review of the manuscript. Referee 2 – line 602: ""Whenever possible, all parameters were analyzed on each plug allowing the identification of statistical and causal relationships between the parameters improving the accuracy of geostatistical predictions" –> this is hard to follow, maybe split the sentence?" Answer: Agreed. We split up the sentence and changed it to "Whenever possible, all parameters were analyzed on each plug. This approach allows the identification of statistical and causal relationships between the parameters and thus, improves the accuracy of geostatistical predictions."

Referee 2 – line 644: "Current studies including detailed petrographic analyses and ICP-MS measurements, aiming to provide a better description and sample classification (Weydt et al., 2020, in prep.)" Answer: We changed the sentence as followed: "Current studies on the reservoir core samples including detailed petrographic analyses and ICP-MS measurements aim to provide a better sample description and classification."

Referee 2 – line 657: "which concept?" Answer: This sentence refers to the statement that "data from the exhumed system in Las Minas can be used as an analogue for modelling the Acoculco geothermal system" in the sentence before. No changes are needed here.

---

## Editor Decision (ED1)

Editorial comments to the revised version of essd-2020-139 "Petrophysical and mechanical rock property database of the Los Humeros and Acoculco geothermal fields (Mexico)" by Leandra M. Weydt et al.

Dear Leandra Weydt and co-authors,

First of all, many thanks for your revision of the manuscript according to the suggestions of the referees. Before finally accepting the manuscript for publication in ESSD, I would like to ask you for some additional adjustments of the manuscript listed below. My source is your Author's response from 12 November 2020. My main requests are to include some of your answers to referee comments in the manuscript. This can avoid similar questions from readers and improve the data description. I have listed my specific comments below.

**Manuscript main text**

**Comment to "Referee 1 – C2 line 329"**

I think that the "second datasheet of the database" is still a little unprecise. Accessing the DOI of the data, I find 3 Excel data tables and one data description pdf. The Excel files are (1) the full database with 9 spreadsheets, (2) a file with petrophysical and rock mechanical properties with 1 spreadsheet and (3) a file with geochemical data in one spreadsheet. To make the correct link to the respective data file and spreadsheet I suggest to include the file (and spreadsheet) in the manuscript. You could further consider to add a citation to the data, e.g. "(Weydt et. al., 2020)"

**Comments to your answers to "Referee 1- C2 line 350"**

Please include a summary of your answer in the manuscript to avoid similar questions by readers

**Comment to "Referee 1 – C2 line 353 and 439":**

Please include your explanations on the effectiveness of saturating samples by leaving them submersed in water in the manuscript. If the referee had this question, we can easily avoid similar question by addressing this topic in the manuscript.

**Comment to "Referee 1 – C3 line 377"**

Please include your comments on the Forchheimer correction in the manuscript

**Comment to "Referee 1 – C3 line 425"**

Please include a summary of your comment in the manuscript

**Comment on "Referee 2, line 29-31 in the authors response" –**

It would be interesting to include your comments on the usability of your database at other places in the discussion. I like your reorganised discussion and think adding a comment on the suitability of the database beyond Los Humeros would complement it.

**References:**

I also found some missing DOIs: please add them as a service for your readers who can directly access the data via the DOI links. Please also make sure that all DOIs in the references section are pre-set with https://doi.org.

**Global comments:**

- Please preset all DOIs with https://doi.org/
- Please change all "dx.doi.org" to "doi.org" (dx.doi.org is the old resolver that is not used anymore)
- Please change all "DOI: 10.xxx" and "doi: 10.xxx" or "doi10.xxx" to https://doi.org/10.xxx and make the links executable

**Comments to individual references (mostly "please add DOIs")**

- Ali and Potter (2012): please write  https://doi.org/10.1190/GEO2011-0282.1
- Békés et al. (2020): please remove the second "101880" after the DOI (this is a typo)
- Calcagno et al. 2018: please correct the DOI syntax from "doi: org/10" to https://doi.org/10"
- Christie (1996): https://doi.org/10.2118/37324-JPT
- Eggertsson et al. (2020) https://doi.org/10.1016/j.jvolgeores.2018.04.008
- Ferriz and Mahood (1984): https://doi.org/10.1029/JB089iB10p08511
- Fuentes-Guzmán etl al (2020): the paper is published, please add the following DOI: http://dx.doi.org/10.18268/BSGM2020v72n3a110520
- Garcia Palomo et al. (2002): I suggest to add this URL: http://www.revistas.unam.mx/index.php/geofisica/article/view/40100
- Guo et al. (1993): https://doi.org/10.1016/0013-7952(93)90056-I
- Hartmann et al. (2005): https://doi.org/10.1016/j.ijrmms.2005.05.015
- Heap et al. (2014): https://doi.org/10.1002/2013JB010521
- ISRM (1988): https://doi.org/10.1016/0148-9062(88)91871-2
- Kummerow and Raab (2015): https://doi.org/10.1016/j.egypro.2015.07.855
- Navelot et al- (2018): please remove the "blank" in the DOI (after "eores"
- Ohnaka (1969): https://doi.org/10.5636/jgg.21.495
- Pola et al. (2014): https://doi.org/10.1016/j.enggeo.2013.11.011
- Rühaak et al. (2015): https://doi.org/10.1016/j.geothermics.2015.08.004
- Rybacki et al. (2015): https://doi.org/10.1016/j.petrol.2016.02.022
- Sanches-Vila et al. (2006): https://doi.org/10.1029/2005RG000169
- Saller and Henderson (1998): https://doi.org/10.1306/1D9BCB01-172D-11D7-8645000102C1865D
- Sass et al., (1971): https://doi.org/10.1029/JB076i026p06376
- Shankland et al. (1997): https://doi.org/10.1029/96JB03389
- Sosa Caballos et al. (2018) https://doi.org/10.1016/j.jvolgeores.2018.06.002
- Vinciguerra et al- (2008): https://doi.org/10.1016/j.ijrmms.2005.05.022
- Vosteen and Schellenschmidt (2003): https://doi.org/10.1016/S1474-7065(03)00069-X
- Wen and Gómez Hernandes (1996)_ https://doi.org/10.1016/S0022-1694(96)80030-8

---

## Author Response (AR2)

**Reply to the editorial comments to the revised version of essd-2020-139 "Petrophysical and mechanical rock property database of the Los Humeros and Acoculco geothermal fields (Mexico)" by Leandra M. Weydt et al.**

Dear Dr. Kirsten Elger,

Thank you very much for processing our manuscript and your detailed proofreading. Please find below our answers to your comments and the modifications carried out in the manuscript.

**1) Comment to available Excel files stored in the TUdatalib data repository**

Answer: As described in the manuscript, we provide our database in two formats: 1) standard Excel file (.xlsx) and 2) as .csv – files. The standard Excel file comprises two spreadsheets (or datasheet): the first one includes all metadata and rock properties and the second one includes all geochemical data. Therefore, we are referring to the "first" or "second datasheet of the database" in the manuscript. The .csv files are needed for users that want to access the data directly via software or other program languages. Thus, each spreadsheet of the database needs to be stored separately as .csv file. Hence, three Excel data tables should be available for the user.

Unfortunately, we recognized that during the update of the database on 11[th] of November, the wrong Excel file (.xlsx) has been uploaded. We now uploaded the correct Excel file and requested to delete the wrong version to our TUdatalib platform service team. The data repository should now include three Excel data tables as described above. We hope this clarifies the confusion.

Since the database comprises more than 140000 entries, we think it is not possible to integrate the spreadsheet into the manuscript or add it as an appendix in form of a pdf-file. Therefore, we like to keep referring to it as a DOI-link, which is already integrated in the abstract, introduction, data availability, and references as requested by the journal at the beginning of the review process.

**2) Comments to further specify the applied methods in the manuscript**

Answer: We agree that providing as much detail as possible of the applied methods is beneficial for the user and increases the reproducibility of this study. However, the measurements conducted in this study were carried out according to internationally recognized testing methods, which are described in detail in the different standards and are developed and verified by numerous norming institutions and committees like the International Society for Rock Mechanics and Rock Engineering (ISRM), American Society for Testing and Materials (ASTM International) or other national standards (e.g. DIN). The methods have been applied worldwide in the past 40-50 years and longer. Therefore, we prefer to keep the manuscript as concise as possible and would like to add only short comments to the respective sections. It is not our aim and not the focus of this study to justify standard methods or to argue which of the applied methods is more suitable. Such a review could easily fill another paper and we would prefer to not distract the reader by adding a half to one-page summary/discussion to the manuscript.

We added the following comments to the revised manuscript:

- Microcracking – p. 12 line 349:

"Microcracking or significant mass losses caused by mineralogical changes or the collapse of clay minerals during heating in the oven were not observed since the majority of the outcrop samples contain no clays and samples affected by hydrothermal or metamorphic processes contain mineral assemblages developed at higher temperatures."

- Differences between porosity measurement methods – p. 12 line 373:

"Variations in particle and bulk density between the different methods applied on the same samples range between 0.3-3% (coefficient of variation) for limestones with porosities smaller than 3 % and 0.5-3.5% for pyroclastic rocks with porosities between 11 and 15%, verifying the different methods and sample saturation procedures as sufficient to obtain data with the needed accuracy."

- Forchheimer correction – p. 12 line 388:

"The recorded flow rates were tested for turbulent fluid flow according to Kushnir et al. (2018) prior to the Klinkenberg correction to ensure laminar fluid flow. A correction after Forchheimer (1901) was not required, since the corrected values were within the error range of the measurement device."

- Provide more information on how the saturated velocities were measured – p. 14 line 432:

"For analyzing the samples at saturated conditions, the samples were stored in degassed and de-ionized water to avoid desaturation. After preparing the device and measurement set-up, the samples were immediately installed between the transducers, and the transmitted signals were recorded until the sample starts to desaturate."

**3) Comment on the usability of the database**

Answer: Thank very much for this comment. In our opinion, the suitability of this database has already been extensively described for general applications independently of the study area in the discussion. Thus, both caldera systems only served as case studies for super-hot unconventional geothermal systems in general as described in the abstract, introduction and project outline. This study aims to provide detailed raw data for various applications with respect to such systems (e.g. feeding calculations for reservoir assessment and 3D geological models with raw data) and to improve the understanding of the relationships between different rock properties and how they are affected by diagenetic or hydrothermal processes, which occur in these geological settings.

As written in the manuscript, the data could be used for comparable geological settings within the TMVB or similar play types worldwide. The TMVB is a quite large and relatively unexplored area when it comes to geothermal exploitation (three of the biggest geothermal reservoirs in Mexico are located there) and there is a big interest to built further power plants. The first example would be Acoculco, which is still a greenfield, but a potential candidate for the development of a geothermal reservoir.

The data from Los Humeros and Las Minas could be used, and already have been used within the GEMex project, for first reservoir assessments and modelling approaches. Furthermore, the carbonatic sequences in the study area and in the TMVB are still an important target for the oil and gas industry and the ore bodies in Las Minas are one of the biggest and most important mines in the country. Regarding other study areas beyond the TMVB, the data can certainly be used for a first assessment in the oil and gas, mining or geothermal industry. The 'suitability' of the data strongly depends on the purpose, the scale (from global to local) and area/geological context of future projects. The detailed sample description, available metadata and geochemical data allows the user to check whether this data meets the requirements of their project or not. Thus, we prefer not to extend the discussion by another section – also with respect to the meanwhile overwhelming length of the manuscript – or to restrict our data to a specific site.

**4) Comments to references and DOI's**

Answer: Thank you very much for this hint. All DOIs were added and updated and all links were activated to increase the accessibility for the reader.